# Human ventromedial prefrontal cortex is necessary for prosocial motivation

**Patricia L. Lockwood** [1,2,3,4,8] ✉, **Jo Cutler** [1,2,3,4,8] ✉, **Daniel Drew**[3,4,5],
**Ayat Abdurahman**[3,4,6], **Deva Sanjeeva Jeyaretna** [5,7], **Matthew A. J. Apps** [1,2,3,4],
**Masud Husain** [3,4,5,7] & **Sanjay G. Manohar** [3,5,7]

Ventromedial prefrontal cortex (vmPFC) is vital for decision-making. Functional neuroimaging links vmPFC to processing rewards and effort, while parallel work suggests vmPFC involvement in prosocial behaviour. However, the necessity of vmPFC for these functions is unknown. Patients with rare focal vmPFC lesions ($n = 25$), patients with lesions elsewhere ($n = 15$) and healthy controls ($n = 40$) chose between rest and exerting effort to earn rewards for themselves or another person. vmPFC damage decreased prosociality across behavioural and computational measures. vmPFC patients earned less, discounted rewards by effort more, and exerted less force when another person benefited, compared to both control groups. Voxel-based lesion mapping revealed dissociations between vmPFC subregions. While medial damage led to antisocial behaviour, lateral damage increased prosocial behaviour relative to patients with damage elsewhere. vmPFC patients also showed reduced effort sensitivity overall, but reward sensitivity was limited to specific subregions. These results reveal multiple causal contributions of vmPFC to prosocial behaviour, effort and reward.

The ventromedial prefrontal cortex (vmPFC) is considered critical for decision-making. Functional neuroimaging studies indicate that vmPFC is key for processing costs such as effort and benefits such as rewards, as well as integrating these to compute value[1–8]. Activity in vmPFC has also been linked with social decisions involving other people[9–12]. However, neuroimaging cannot reveal causal evidence for cognitive processes. Moreover, most lesion studies of vmPFC are either case studies or contain groups of fewer than ten patients[13–21]. Given typical individual variability in processing social behaviour[22], effort[23–25] and reward[26], larger samples are necessary to reach robust conclusions. Small group studies also prevent lesion-mapping approaches that can differentiate between different subregions within an area of interest. This is particularly important for vmPFC, a heterogeneous region

comprising several distinct cytoarchitectonic zones[27]. Here we tested the effects of damage to vmPFC on social behaviour in a paradigm that independently manipulated effort and reward for oneself and another person. We compared a large sample of patients with rare focal vmPFC lesions ($n = 25$) to two carefully matched control groups to reveal the involvement of specific vmPFC subregions in prosocial behaviour, effort and reward.

Every day we make choices between effort and reward, such as whether to go to the gym or stay at home, or between helping a colleague and watching TV. Effort is aversive. If two courses of action are associated with different amounts of effort but the same reward, people will choose the less effortful option[28–32]. A lack of willingness to put in effort has important implications for everyday behaviour and

[1]Centre for Human Brain Health, School of Psychology, University of Birmingham, Birmingham, UK. [2]Institute for Mental Health, School of Psychology, University of Birmingham, Birmingham, UK. [3]Department of Experimental Psychology, University of Oxford, Oxford, UK. [4]Wellcome Centre for Integrative Neuroimaging, University of Oxford, Oxford, UK. [5]Nuffield Department of Clinical Neurosciences, University of Oxford, Oxford, UK. [6]Department of Psychology, University of Cambridge, Cambridge, UK. [7]Department of Neurology, John Radcliffe Hospital, Oxford, UK. [8]These authors contributed equally: Patricia L. Lockwood, Jo Cutler. ✉e-mail: P.L.Lockwood@bham.ac.uk; J.L.Cutler@bham.ac.uk

characterizes apathy, a clinical condition of reduced motivation that is common across multiple psychiatric and neurological disorders[33,34]. It is therefore essential to understand the neural mechanisms of how costs such as effort and benefits such as rewards are integrated into value-guided behaviour.

While many of our choices are made to benefit ourselves, we also often decide whether to exert effort to provide benefits for other people. These 'prosocial' behaviours are crucial for human success as they allow us to cooperate and address global challenges, and have important benefits for the individual, society and the recipient[35,36]. While our tendency to donate rewards to other people is often studied as prosocial behaviour, prosocial motivation is the tendency to voluntarily perform effortful actions to benefit others. Most prosocial acts are indeed effortful, from holding open doors to helping colleagues with their work, but our understanding of how effort impacts prosociality remains limited.

Previous research has shown that young adults are less willing to put in effort to help other people than they are to help themselves, particularly when the effort costs are high[22,24,37,38]. They also exert less force into prosocial acts, whereas older adults become relatively more prosocial and exert equal force[37]. Other work has tied age-related differences in prosocial behaviour to vmPFC maturation[39], suggesting a possible neural basis to these differences. Meta-analyses of neuroimaging studies have identified vmPFC as a crucial region for prosocial behaviour across studies[9], and vmPFC activity has been found when participants observe another person receiving rewards[40,41]. However, vmPFC is susceptible to high signal dropout in MR imaging[9,42], which can affect the ability to localize function in this area.

A smaller number of studies have examined prosocial and moral choices, or effort and reward processing, in patients who have damage to vmPFC[13–16,43–45]. Lesion studies offer valuable insights into causal functions of brain regions and have associated vmPFC with processes including value-based decision-making and social cognition[46]. Seminal case studies have suggested that damage to this area profoundly disrupts social behaviour[13,43]. Patients with vmPFC damage may also be less generous and less trustworthy, experience less guilt, and reject more unfair offers in economic games[14,17]. In moral dilemmas, vmPFC lesions have been linked to abnormally utilitarian preferences and impaired judgements of harmful intentions[15,16]. In contrast, other work has linked vmPFC damage with increased willingness to cooperate with a group[44]. However, an important consideration when assessing vmPFC involvement in social behaviour is whether studies use real rather than hypothetical social situations, given that distinct prefrontal cortex regions may play distinct roles in these processes[45]. Another is that economic games cannot dissociate whether any differences in prosocial behaviour are due to how rewards are valued in general or specifically due to the value of donating rewards to other people, since the cost for self and reward for other are both financial.

One possible explanation for conflicting findings is that vmPFC is a heterogeneous region comprising several distinct cytoarchitectonic zones[27,47–49]. Indeed, the term 'vmPFC' is not an anatomical but a functional characterization, and further understanding of the contribution of anatomical subregions within this heterogenous zone is crucial. While different characterizations of vmPFC subregions have been proposed[47,49], one prominent distinction is between medial and lateral areas[47,48]. The most medial areas include area 14 on the medial wall (comprising 14m, 14a and 14c), whereas more lateral areas include those on the orbital gyrus or area 13 (comprising 13a, 13b and 13m). These distinctions could be key for understanding distinct functions[50–53]. So far, different functional contributions have been hypothesized, including unique roles in representing positive and negative valence[53–56], primary and secondary reinforcement[57], or processing identity[58,59], which is crucial for separating self and other. However, since focal vmPFC damage is so rare, most lesion studies are case studies or involve fewer than ten patients[13–21]. Larger samples may permit lesion-mapping approaches

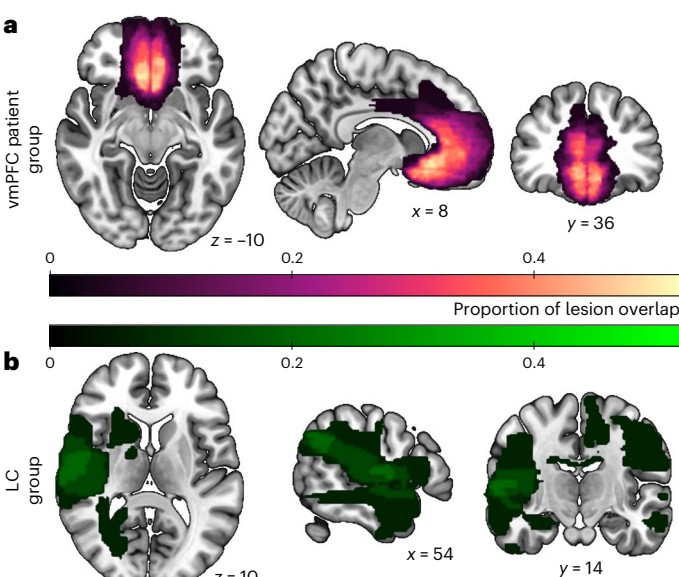

**Fig. 1 | Lesion locations for the vmPFC lesion patients and LC group. a**, Patients in the vmPFC group (n = 25) had focal damage to vmPFC, mostly caused by subarachnoid haemorrhage, with lesions spanning lateral portions (area 13) of bilateral vmPFC and subregions on the medial surface (areas 14, 25 and 32). **b**, Patients in the LC group (n = 15) also had damage mostly caused by subarachnoid haemorrhage but to regions outside of vmPFC, and also not covering dorsal mPFC or ACC (Methods).

that can differentiate between these different vmPFC subregions within the lesioned area. Lesion mapping can also reveal paradoxical positive and negative associations between behaviour and damage in close but distinct areas[60].

Here, we tested the causal role of vmPFC in processing prosocial behaviours in the context of effort and reward by comparing a large group of patients with rare focal lesions in vmPFC (n = 25; Fig. 1a) with two control groups: patients with lesions elsewhere (lesion controls (LC); n = 15; Fig. 1b) and age- and gender-matched participants with no brain damage (healthy controls (HC); n = 40). The participants completed a decision-making task[24], choosing on each trial whether to rest or accept an offer to 'work'—exert physical effort (30–70% of their maximum grip strength) for financial reward (Fig. 2a). Half of the trials were self-benefiting, as the rewards were for the participants themselves (self). Importantly, the other trials created a carefully matched prosocial condition, identical in all aspects apart from the fact that the rewards the participant could earn were for an anonymous other person (other; Methods and Fig. 2b). We compared the groups on the amount of reward earned overall for each recipient, choices to exert effort and force applied. To precisely quantify how effort and reward are integrated into choices, we developed a computational neurology approach, fitting multiple computational models to the data and incorporating the resulting parameters in lesion–symptom mapping[24].

We show converging evidence that vmPFC damage decreases prosociality, compared with control participants both with and without lesions. When another person would benefit, vmPFC patients earned less, were more reluctant to exert effort and physically exerted less force compared to both control groups. This decreased willingness to choose effortful actions that help another person was most strongly associated with lesions in a particular subregion of vmPFC on the medial surface through voxel-based lesion–symptom mapping (VLSM) analysis. However, this analysis also identified a more lateral vmPFC subregion where damage was associated with relatively increased prosociality, with rewards for others devalued less relative to rewards for self. Lesions in vmPFC additionally reduced sensitivity to effort but not reward overall. However, the VLSM analysis also identified specific

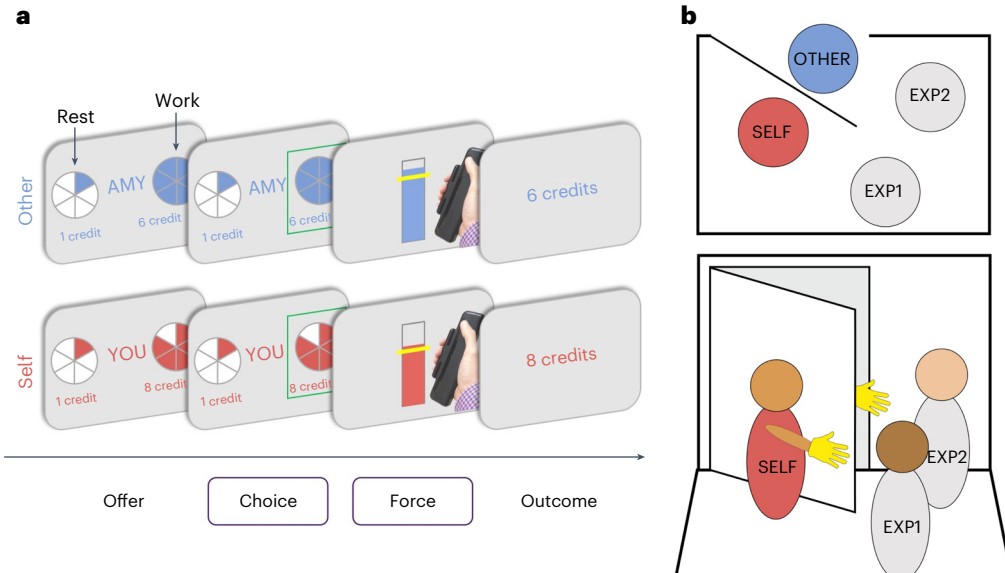

**Fig. 2 | Effort-based decision-making task with rewards for self or other.**
**a**, Before any further instructions, the participants squeezed as hard as they could to measure their maximum voluntary contraction (MVC) on a handheld dynamometer to threshold each effort level to their strength. After thresholding and practice, the participants chose on each trial between a 'rest' option, which required no effort (0% MVC, one segment of the pie chart) for a low reward of one credit, and a 'work' option, which required more effort (30–70% MVC, two to six pie chart segments) for more reward (two to ten credits). The reward available and the effort required were manipulated independently. After making their choice, the participants had to exert the required force (shown by the yellow line) for at least 1 s of a 3 s window to receive the reward. Visual feedback on the amount of force used was displayed on the screen. The participants then saw

the outcome, depending on the offer they had chosen and whether they were successful. If they did not meet the required force level, '0 credits' was displayed. Crucially, on self trials, the participants made the choice, exerted the effort and received the reward themselves, whereas on other trials ('AMY' in this example), the participants made the choice and exerted the effort, but the other participant received the reward. **b**, The participants were designated as 'Player 1' (self) and told that their decisions impacted another player, 'Player 2' (other), who they met at the beginning of the testing session with their identity obscured (to control for influences of identity or reciprocity). The name used in the task was gender-matched to the participant. The procedure involved four people: two experimenters, EXP1 and EXP2, and two participants, self and other.

subregions of vmPFC where damage did decrease reward sensitivity. Together our findings provide evidence of multiple, specific, causal roles of vmPFC in prosocial behaviours, effort and reward processing, with different subregions contributing distinct functions.

## Results

To establish the impact of vmPFC damage on prosocial behaviour, effort and reward processing, we compared 25 patients with vmPFC damage (mean age 56.44 years; 14 females) to two control groups: 15 LC patients with damage to areas outside vmPFC (mean age 56.00 years; 10 females) and 40 HC participants with no brain damage (mean age 60.00 years; 23 females). All participants completed an effort-based decision-making task[24] that independently manipulated effort, reward and recipient. The participants chose between a low-effort, low-reward 'rest' option and a high-effort, high-reward 'work' option on each trial. On half of the trials, the reward was for themselves (self), and on the other half, the reward was for an anonymous other person (other; Methods and Fig. 2). The three groups were carefully matched, with no differences in gender, age, cognitive ability or level of apathy. The two lesion groups also did not differ from each other in education or depression (Methods and Supplementary Table 1). However, the vmPFC and LC groups had lower levels of education than the HC participants (Supplementary Table 1). We therefore repeated all our behavioural analysis controlling for education as a covariate. Controlling for education did not change any of our key results or inferences regarding group differences in prosocial behaviour (Supplementary Tables 2–7).

### Patients with vmPFC damage earn less for the other person
Our first analysis examined how many credits participants earned for themselves and for the other anonymous participant, which translated into bonus money for each recipient. We tested whether the

number of credits earned differed between the three groups (vmPFC, HC and LC) using a generalized linear mixed-effects model (GLMM) (Methods and Supplementary Table 2). Patients with vmPFC damage earned fewer credits for the other relative to self than both control groups (group × recipient interaction; vmPFC versus HC odds ratio$_{150}$ (OR) (95% confidence interval) = 0.88 (0.81, 0.95); $P$ = 0.003; vmPFC versus LC OR$_{150}$ = 0.87 (0.78, 0.97); $P$ = 0.010; Fig. 3a). Post hoc tests revealed that these interactions were driven by the vmPFC group earning significantly fewer credits for the other person than both HC participants (ratio = 0.73; s.e. = 0.07; $Z$ = −3.13; $P$ = 0.002) and patients with damage elsewhere (ratio = 0.76; s.e. = 0.10; $Z$ = −2.18; $P$ = 0.030). In contrast, vmPFC patients and LC patients did not statistically differ in the amount they earned for themselves, with Bayesian evidence supporting the null (ratio = 1.00; s.e. = 0.13; $Z$ = 0.02; $P$ = 0.99; Bayes factor (BF$_{01}$) = 3.05; Methods). Patients with vmPFC damage also did not significantly differ from HC participants (ratio = 0.95; s.e. = 0.09; $Z$ = −0.54; $P$ = 0.59; BF$_{01}$ = 2.10). The two control groups did not significantly differ in the number of credits earned for either recipient (other ratio = 1.04; s.e. = 0.12; $Z$ = 0.31; $P$ = 0.75; BF$_{01}$ = 3.17; self ratio = 1.06; s.e. = 0.13; $Z$ = 0.47; $P$ = 0.64; BF$_{01}$ = 2.87). Together, these results show that damage, specifically to vmPFC, decreased rewards earned for the other person.

### Damage to vmPFC decreases prosocial behaviour
To examine what was driving the group difference in the broad measure of credits earned, we first considered participants' willingness to choose the high-effort work option over rest. Using a binomial GLMM (Methods), we tested whether vmPFC lesions affected prosociality (recipient × group interaction) in choices. Analysis of choices on each trial enabled us to test group differences in sensitivity to rewards (reward levels 2–6: two, four, six, eight or ten credits) and effort

(effort levels 2–6: 30%, 40%, 50%, 60% or 70% of maximum, squared to match the computational model; Methods). We also considered the possibility of higher-order interactions by starting with a maximal model containing fixed effects of and all interactions (up to four-way) between effort, reward, recipient and group, as well as per-participant random effects and all interactions (up to three-way) between effort, reward and recipient. Data-driven model reduction suggested that no interaction terms between more than two variables significantly improved the model (Methods).

Prosociality decreased with vmPFC damage (Supplementary Table 3). vmPFC patients chose to work to benefit the other person less than both control groups (vmPFC versus HC OR = 0.33; s.e. = 0.17; $Z = -2.11$; $P = 0.034$; vmPFC versus LC OR = 0.23; s.e. = 0.15; $Z = -2.19$; $P = 0.029$; Fig. 3b). In contrast, the vmPFC group was equally willing to exert effort to benefit themselves compared to both control groups (vmPFC versus HC OR = 0.91; s.e. = 0.51; $Z = -0.17$; $P = 0.87$; $BF_{01} = 3.57$; vmPFC versus LC OR = 1.01; s.e. = 0.72; $Z = 0.01$; $P = 0.99$; $BF_{01} = 3.13$). Finally, the two control groups did not significantly differ in willingness to exert effort for self or other, although these contrasts were not sensitive enough to support evidence of no difference (self OR = 1.11; s.e. = 0.73; $Z = 0.16$; $P = 0.88$; $BF_{01} = 2.95$; other OR = 0.70; s.e. = 0.43; $Z = -0.58$; $P = 0.56$; $BF_{01} = 2.56$).

### vmPFC lesions decrease effort sensitivity

Analysis of effort sensitivity also revealed a significant difference between HC participants and vmPFC patients (group × effort interaction OR = 0.44 (0.28, 0.69); $P < 0.001$; Fig. 3c). HC participants accepted a larger number of work offers at the lower effort levels (level 2 mean = 0.92), but the willingness to work dropped dramatically when more effort was required (level 6 mean = 0.64). This meant that effort had a negative effect on choices for HC participants (effort effect (confidence level) = −1.73 (−2.00, −1.00)). In contrast, while vmPFC patients were also less willing to work as effort increased, the effect of effort on choices was significantly less negative (effect = −0.91 (−1.32, -0.51)). This group difference was due to vmPFC patients accepting relatively fewer work offers when the effort required was minimal (level 2 mean = 0.81) but showing a similar willingness to work at the highest effort level (level 6 mean = 0.63; Fig. 3c). Follow-up analyses showed reduced effort sensitivity in vmPFC patients compared with HC participants for both recipients (self OR = 0.48 (0.29, 0.78); $P = 0.003$; other OR = 0.42 (0.26, 0.68); $P < 0.001$). Effort also negatively affected LC patients' willingness to work, but this did not significantly differ from the other groups (LC versus vmPFC GLMM group × effort interaction OR = 0.73 (0.42, 1.27); $P = 0.27$; LC versus HC post hoc interaction = −0.50; s.e. = 0.27; $Z = −1.87$; $P = 0.061$).

Analysis of sensitivity to reward showed that participants in all three groups chose to work more when a larger reward was available (vmPFC GLMM OR = 6.91 (4.70, 10.15); $P < 0.001$; Fig. 3d), particularly when the reward was for themselves rather than for another person (recipient × reward interaction OR = 1.71 (1.46, 2.01); $P < 0.001$). However, there was no evidence of group differences in reward processing (Fig. 3d). The group × reward interaction was not selected as a term in the GLMM, and there was strong evidence against adding it to the best-fitting model (Δ deviance $\chi^2_2 = 2.74$; $P = 0.25$; difference in Bayesian information criterion (ΔBIC) = 15.98; $BF_{01} = 2,950.76$).

### Reward devaluation by effort depends on the recipient

So far, we have shown that vmPFC damage has a detrimental impact on prosocial decision-making and that this is at least partially due to vmPFC patients' lower willingness to exert prosocial effort, compared with both control groups. We next fitted computational models of effort discounting to participants' choice behaviour to precisely quantify how the effort required was integrated with the reward available to determine choices for self or other. Importantly, these models separate the extent to which effort devalues the reward (the discounting

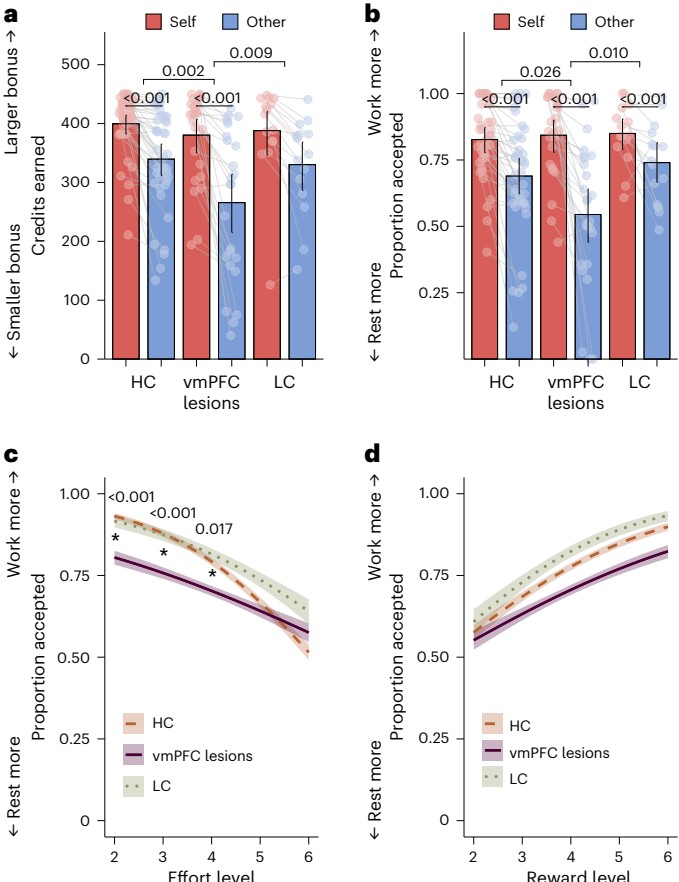

**Fig. 3 | Damage to vmPFC decreases prosociality and effort sensitivity. a,b,** vmPFC lesions decreased willingness to help others, measured as earning fewer credits for the other player but not for themselves (**a**) and accepting fewer high-effort high-reward 'work' offers for the other but not for themselves (**b**). The data are presented as mean values ± s.e.m. The dots show individual data points for each participant ($n = 80$). Significance lines between groups represent significant interactions between the recipient (self or other) and group in the (G)LMM (Results and Supplementary Tables 2 and 3). Significance lines between self and other represent differences within each group in post hoc comparisons (two-sided, all $P < 0.001$ uncorrected; follow-ups of significant GLMM interactions; Results). Significance levels for comparisons between groups for each recipient are not labelled but show that the significant recipient × group interactions are driven by vmPFC patients earning significantly fewer credits and accepting significantly fewer work offers for the other person than both control groups (all $P < 0.03$ uncorrected; follow-ups of significant GLMM interactions; Results). In contrast, there were no significant differences between vmPFC patients and the control groups in self-benefiting decision-making (all $P > 0.59$), and there was Bayesian evidence of no credible difference in credits earned (versus LC) or choices to work for self-benefiting rewards ($BF_{01} > 3.05$). **c,** The effect of effort on decision-making differed between vmPFC patients and participants without brain damage (estimates from a GLMM of binary choices between work and rest; group × effort interaction OR = 0.44 (0.28, 0.69); $P < 0.001$). This group difference in effort sensitivity was driven by vmPFC patients' reduced willingness to accept the work offer at the lower effort levels compared with HC participants. Significance levels are shown for differences in post hoc comparisons at each effort level (two-sided, all $P < 0.02$ uncorrected; follow-ups of significant GLMM interactions). The data are presented as generalized linear model (GLM)-estimated conditional means ± 95% confidence intervals. **d,** In contrast, there were no significant group differences in how reward affected decisions. All three groups were more willing to exert effort as the reward available increased, and there was no evidence that this sensitivity to reward was significantly different in the vmPFC group. The data are presented as GLM-estimated conditional means ± 95% confidence intervals.

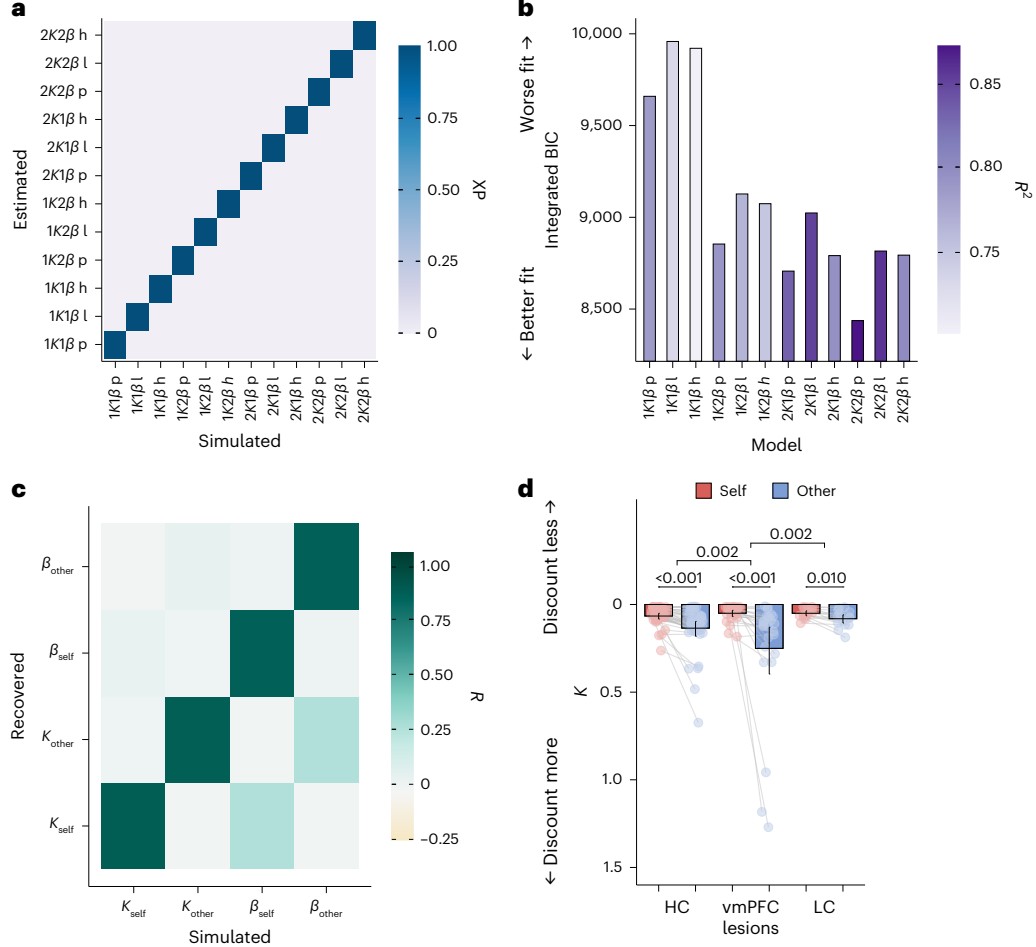

**Fig. 4 | A computational neurology approach reveals that vmPFC damage increases discounting of rewards for others by effort. a**, The results of the model identifiability analysis show a strong diagonal across the average XP confusion matrix, meaning that the models of interest can be accurately identified. We simulated ten datasets each with 100 artificial agents, using each of the 12 models. Parameters were drawn at random from a flat distribution between the upper and lower bounds used when fitting the model to the participant data ($0 < K < 1.5$, $0 < \beta < 4$). We then fit each dataset using the same model fitting and comparison procedure applied to the participant data and plotted the confusion matrix from the XP averaged across the ten runs. h, hyperbolic; l, linear; p, parabolic. **b**, Model comparison of the 12 candidate models shows that the $2K2\beta$ parabolic model best fits the participant data on the basis of multiple fit metrics. This model had a parabolic discounting function and contained separate discounting and choice consistency parameters for self and other. It had the highest XP (XP = 1) and the lowest integrated BIC, and

it explained the highest proportion of variance in choices ($R^2 = 0.87$). **c**, Strong parameter recovery for the $2K2\beta$ model is shown by the high correlations on the diagonal between simulated and recovered parameter values and the low off-diagonal correlations. Simulations for parameter recovery used a grid of values for $K$ (0, 0.3, 0.6, 0.9, 1.2 and 1.5) and $\beta$ (0, 1, 2, 3 and 4), with added noise drawn from a normal distribution × 0.05, to cover the full range of possible values. **d**, Comparing discounting ($K$) parameters from the winning model between recipients and groups shows that vmPFC damage increases the discounting of prosocial rewards. The data are presented as mean values ± s.e.m. The dots show individual data points for each participant ($n = 80$). Significance lines between groups represent significant interactions between the recipient (self or other) and group in the GLMM (Results and Supplementary Table 4). Significance lines between self and other represent differences within each group in post hoc comparisons (two-sided, all $P < 0.01$ uncorrected; follow-ups of significant GLMM interactions).

$K$ parameter) from choice consistency (the inverse stochasticity $\beta$ parameter) and therefore allow us to test which of these parameters differ between recipients and groups.

We fitted and compared multiple models to participants' choice behaviour using a hierarchical expectation maximization approach[61,62] (Methods). These models differed in whether there were separate discounting and consistency parameters for self and other, or whether combined parameters across recipients captured choice behaviour. We also compared different discounting functions (linear, hyperbolic and parabolic) that represent the effect of the effort required on discounting the available reward[24,37,38] (Fig. 4a,b and Supplementary Methods). The best model, $2K2\beta$, had a parabolic discount function and a separate parameter in each recipient condition for both the discounting ($K_{self}$ and $K_{other}$) and consistency parameters ($\beta_{self}$ and $\beta_{other}$).

This model showed the best fit through having the lowest integrated BIC (Fig. 4b) and an exceedance probability (XP) of 1. It also had the largest variance in choices explained ($R^2 = 0.87$) and the highest estimated model frequency (EF = 0.54). Importantly, this was also the case for each group separately (HC: XP = 1.00, EF = 0.51, $R^2 = 0.88$; vmPFC: XP = 0.98, EF = 0.55, $R^2 = 0.87$; LC: XP = 0.98, EF = 0.63, $R^2 = 0.87$), and between-group comparisons of model fit showed high probabilities (>0.98) that the model frequencies were the same across groups (Supplementary Methods). The robustness of the model comparison procedure and the winning model was established using simulated data (Methods). Model identifiability results demonstrated perfect identifiability (Fig. 4a). The simulations also showed very good recovery of all parameters in the $2K2\beta$ winning model ($K_{self} = 0.86$, $K_{other} = 0.86$, $\beta_{self} = 0.85$, $\beta_{other} = 0.84$; Fig. 4c).

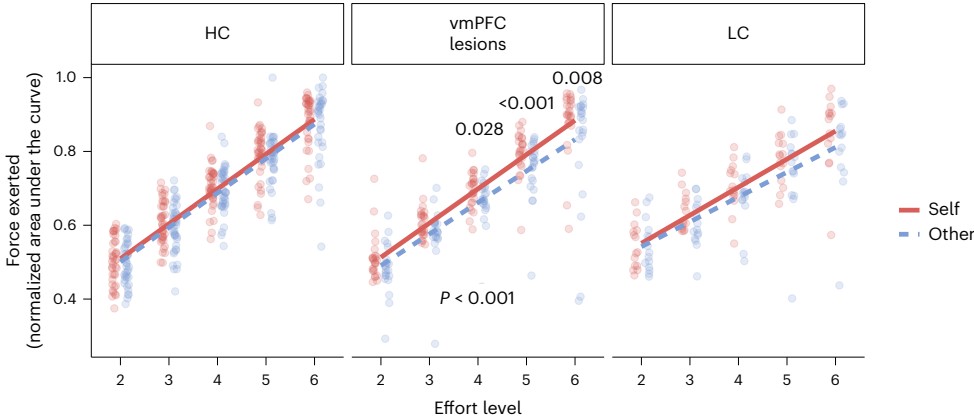

**Fig. 5 | Patients with vmPFC lesions exert less force for others than for themselves, particularly when a large amount of effort is required.** Patients with vmPFC damage exerted less force to obtain a reward for the other person than for themselves (post hoc contrast = 0.04; s.e. = 0.01; $Z$ = 3.62; $P$ < 0.001), whereas this was not the case for HC participants or LC patients (HC: post hoc contrast = 0.01; s.e. = 0.01; $Z$ = 1.32; $P$ = 0.19; LMM group × recipient interaction $\beta_{8638}$ = −0.09 (−0.18, -0.01); $P$ = 0.036; LC: post hoc contrast = 0.02; s.e. = 0.01; $Z$ = 1.70; $P$ = 0.09; LMM group × recipient interaction $\beta_{8638}$ = −0.05 (−0.16, 0.05); $P$ = 0.33). The difference between vmPFC patients and HC participants

in energizing actions to benefit others, relative to self, was particularly large when the level of effort required was greatest (LMM group × recipient × effort interaction $\beta_{8638}$ = −0.03 (−0.05, 0.00); $P$ = 0.037) and when the reward available was smallest (LMM group × recipient × effort × reward interaction $\beta_{8638}$ = 0.05 (0.03, 0.08); $P$ < 0.001; Supplementary Fig. 2). These interactions were not significant when comparing the vmPFC and LC groups (Supplementary Table 6). Significance is shown under the lines for the self-versus-other post hoc contrast and above the lines for post hoc comparisons at each effort level (two-sided, all $P$ < 0.03 uncorrected; follow-ups of significant GLMM interactions).

## vmPFC lesions increase the discounting of prosocial rewards

After establishing that the $2K2\beta$ model was robust and best fit the data, we assessed whether the $K$ parameters differed between groups with a GLMM (Methods). Damage to vmPFC decreased prosociality compared with both control groups (group × recipient interaction vmPFC versus LC OR = 1.46 (1.15, 1.85); $P$ = 0.002; vmPFC versus HC OR = 1.36 (1.12, 1.65); $P$ = 0.002; Fig. 4d and Supplementary Table 4). Patients with vmPFC lesions specifically showed higher discounting of rewards for another person, whereas effort discounting for self was the same regardless of lesion location (vmPFC versus LC: other ratio = 2.18; s.e. = 0.73; $Z$ = 2.31; $P$ = 0.021; self ratio = 1.02; s.e. = 0.34; $Z$ = 0.07; $P$ = 0.94; BF$_{01}$ = 3.16). In contrast, levels of prosociality did not differ between the two control groups (post hoc group × recipient interaction = 0.87; s.e. = 0.19; $Z$ = −0.65; $P$ = 0.52), further supporting the specificity of vmPFC damage. Finally, all groups showed higher effort discounting of rewards for other than for self (Fig. 4d), indicating that the bias towards helping oneself occurs across HC participants and patients, regardless of lesion location.

We used the same approach to examine whether vmPFC damage affected decision consistency (the inverse stochasticity $\beta$ parameter). Across all groups, participants made less consistent decisions in the prosocial than in the self-benefiting condition (all $P$ < 0.007; Supplementary Fig. 1). However, there was no evidence that this self-versus-other difference was significantly affected by vmPFC damage (for LMM group × recipient interactions, all $P$ > 0.18) or that overall levels of decision consistency differed between groups (all $P$ > 0.12; Supplementary Table 5).

## Patients with vmPFC damage exert less force to help others

Decreased prosocial decisions in vmPFC patients at least partially explain why this group earned fewer credits for the other person than both control groups. Our final behavioural analysis considered how much force participants exerted after choosing to exert effort for themselves or for the other person. As with the choice data, we modelled force (normalized area under the curve; Methods) by reducing a maximal LMM that contained fixed effects and all interactions (up to four-way) between effort, reward, recipient and group, as well as random effects and all interactions (up to three-way) between effort, reward and recipient (see Supplementary Table 6 for the full results).

Patients with vmPFC damage exerted less force for the other person, relative to self, overall than HC participants (group × recipient interaction b$_{8638}$ = −0.014 (−0.028, −0.001); $P$ = 0.036; Fig. 5), particularly at higher effort levels (group × recipient × effort interaction $\beta_{8638}$ = −0.004 (−0.008, 0.000); $P$ = 0.037) and when the reward available was lower (group × recipient × effort × reward interaction $\beta_{8638}$ = 0.008 (0.004, 0.012); $P$ < 0.001; Supplementary Fig. 2). Lower exerted force for others is consistent with vmPFC patients exhibiting 'superficial prosociality', sometimes deciding to benefit another person but then not energizing actions enough to do so. The finding that sensitivity to effort and reward were important in determining the force exerted is also evidence against the idea that vmPFC damage led to inattention to the details of the work offer.

We then directly compared success rates between the two groups to test whether underexertion of force on prosocial trials led to less reward being delivered to the other person, rather than an overenergization of action for self. We ran an LMM predicting average success rates in each group for self and other (Supplementary Table 7). The results revealed that patients with vmPFC lesions successfully exerted the required effort more often for self (95.05% success rate) than for others (87.29%; contrast = 0.08; s.e. = 0.02; $t$ = 3.17; $P$ = 0.002). In contrast, HC participants had similar success rates for self (97.57%) and others (96.91%; contrast = 0.01; s.e. = 0.02; $t$ = 0.35; $P$ = 0.73; BF$_{01}$ = 2.95; LMM group × recipient interaction $\beta_{150}$ = −0.319 (−0.592, −0.045); $P$ = 0.023). The high success rate that vmPFC patients had when rewards were for themselves (95.05%) shows that they were perfectly capable of energizing actions, suggesting that a lack of prosocial motivation caused higher rates of failure for others.

In summary, multiple behavioural measures and computational parameters revealed that damage to vmPFC specifically reduced willingness to exert effort to help another person, compared to controls both with and without lesions. Lesions in vmPFC also led to a lower willingness to energize actions, even after deciding to help others.

## Dissociable effects of medial versus lateral vmPFC damage

Finally, we used VLSM to examine whether specific subregions within the heterogenous vmPFC were associated with prosocial behaviour, effort sensitivity and reward sensitivity. The VLSM analysis identifies voxels where patients with damage at that voxel differ from patients

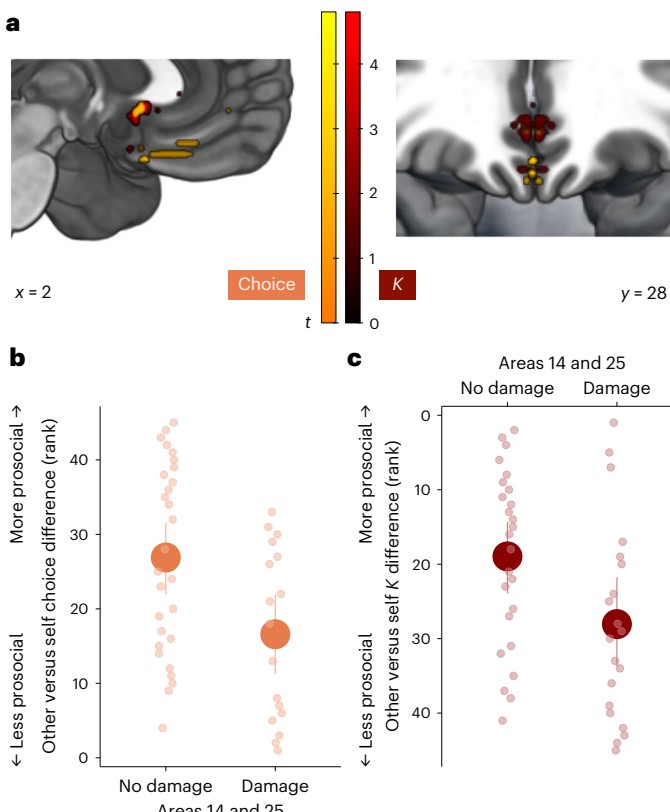

**Fig. 6 | Damage to medial portions of vmPFC specifically is associated with reduced prosocial behaviour. a**, Permutation-based, whole-brain, non-parametric VLSM reveals subregions of vmPFC on the medial surface where damage is associated with reduced prosociality. This effect was found for both the self–other difference in choices to accept the work over the rest option (orange) and the self–other difference in devaluing rewards by effort (*K* parameters; red). Specifically, the recipient effect on choices was associated with damage to area 14 (peak 0, 30, –22; Supplementary Fig. 5) and area 25 (sgACC; peak ±2, 16, –6). The recipient effect on discounting *K* parameters was also associated with damage to area 14 (peak ±2, 14, –20) and area 25 (peak ±2, 14, –8). **b,c**, Plotting the (ranked) effect of the recipient on choices (**b**) and *K* parameters (**c**) separately for participants with damage or no damage in the areas identified by the relevant VLSM analysis reveals that damage decreases prosociality. The data are presented as mean values ± s.e.m. The dots show individual data points for each participant (*n* = 40). To further interpret these differences, we also plotted choices and *K* parameters for self and other separately by damage to the medial vmPFC regions (Supplementary Fig. 6).

with damage elsewhere in terms of self–other differentiation in (1) discounting *K* parameters and (2) choices, as well as how the effort required and reward available determined choices (Methods). Importantly, the overall extent of damage (total lesion size) was not significantly associated with any of these measures (Supplementary Fig. 3). VLSM examines whether each voxel's lesion predicts a person's behaviour by constructing a map of the *t* statistics[60]. We only included voxels where at least five patients had damage[63] (Supplementary Fig. 4). We used the FMRIB software library (FSL)[64] to run a permutation-based VLSM with threshold-free cluster enhancement (TFCE)[65,66] and report significance at *P* < 0.0125 (*P* < 0.05 Bonferroni-corrected across the four behavioural regressors; Methods).

We first ran VLSM analyses on prosociality (the self-versus-other differences in discounting *K* parameters and choices) including all 45 patients, 25 with vmPFC damage and 20 with damage elsewhere (15 in the LC group and 5 additional patients excluded from the group comparisons; Methods). This revealed specific areas on the medial wall of vmPFC where damage was associated with decreased prosociality

(Fig. 6a). As vmPFC patients were generally less willing to help someone else than to help themselves, this effect showed where damage was associated with even less willingness to help the other person relative to oneself. Lesions in regions putatively in area 14 (ref. 47) (Supplementary Fig. 5) and area 25 (ref. 49) (subgenual anterior cingulate cortex (sgACC)) decreased willingness to help on both measures of choices (Fig. 6b) and discounting *K* parameters (Fig. 6c). To further interpret these findings, which are based on self–other differences, we examined whether the damage in these regions was associated with differences specifically for self, other or both recipients. Plotting choices and *K* parameters for each recipient separately showed that damage to these medial regions was associated with both a lower willingness to help others and a higher willingness to exert effort for self-benefiting rewards (Supplementary Fig. 6).

In a second VLSM analysis, we further focused on vmPFC subregions by including only patients in the vmPFC group. This revealed a distinct, lateral portion of vmPFC, putatively in area 13 (ref. 47) (Supplementary Fig. 5), where damage was associated with relatively increased prosociality, compared with damage to other vmPFC subregions, in both choices and effort discounting (*K* parameters; Fig. 7). To interpret these findings, we again plotted the *K* parameters for each recipient

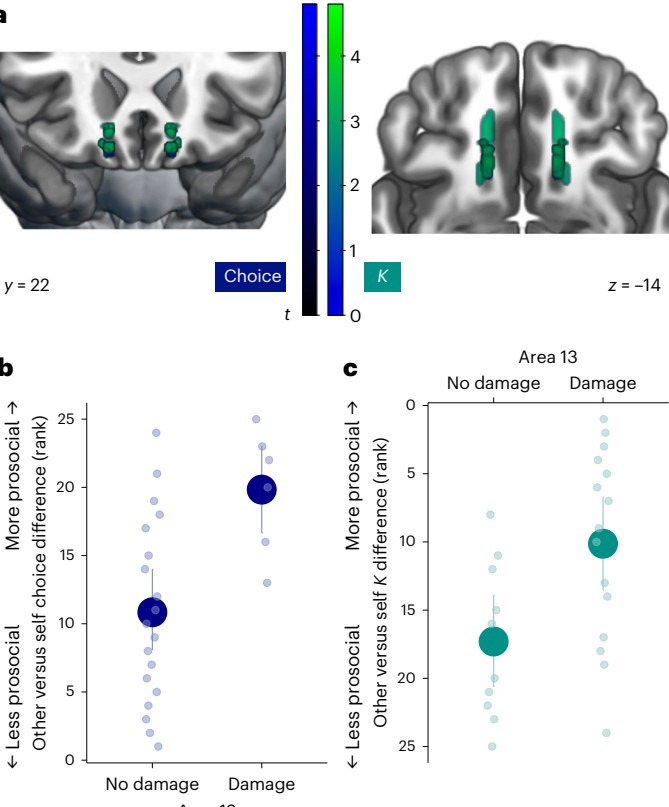

**Fig. 7 | Lateral vmPFC damage is associated with relatively increased prosocial behaviour within the vmPFC patient group. a**, Permutation-based, whole-brain, non-parametric VLSM shows that damage to a lateral portion of vmPFC is associated with relatively increased prosociality (see Supplementary Fig. 7 for comparison with the average behaviour of HC participants). An overlapping region in area 13 was identified in both the VLSM of the self–other difference in choices (dark blue; peak ±12, 20, –22) and how rewards were devalued by effort (*K* parameters; teal; ±14, 20, –12; Supplementary Fig. 5). **b,c**, Plotting the (ranked) self–other difference by whether participants had damage in the identified region shows the relatively increased prosociality in both choices (**b**) and *K* parameters (**c**), compared with patients with damage elsewhere in vmPFC. The data are presented as mean values ± s.e.m. The dots show individual data points for each participant (*n* = 25). To further interpret these differences, we also plotted *K*self and *K*other separately (Supplementary Fig. 7).

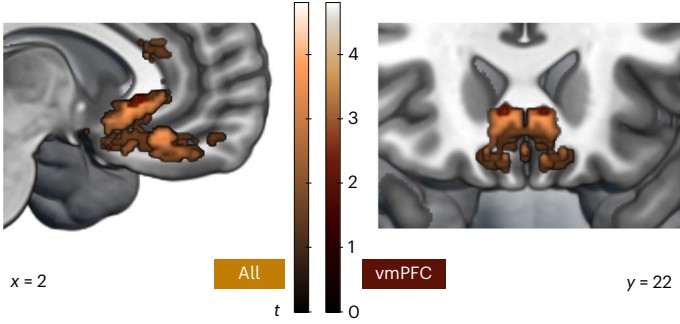

**Fig. 8 | Damage to medial and lateral portions of vmPFC decreases sensitivity to reward.** Permutation-based, whole-brain, non-parametric VLSM shows portions of vmPFC spanning medial and more lateral areas where damage is associated with reduced reward sensitivity. More damage in these subregions, areas 14, 13 and 32 and sgACC, predicted a less positive effect of reward on choices in both the analysis of all patients (light brown; peak 6, 26, −2) and the analysis of the vmPFC group only (dark brown; peak ±6, 24, −2).

separately by whether participants had damage in this vmPFC subregion (Supplementary Fig. 7). This showed that damage to area 13 was associated with lower values of $K_{other}$, and therefore relatively greater willingness to exert effort for prosocial rewards, but relatively higher values of $K_{self}$, meaning devaluing self-benefiting rewards to a greater extent when effort was required.

### Medial and lateral vmPFC is linked to reward sensitivity

Finally, we used VLSM to examine whether any subregions of vmPFC were significantly associated with effort sensitivity or reward sensitivity. There were no areas where damage was significantly associated with the effect of effort on choices at our thresholding criteria (permutation-based TFCE $P < 0.0125$). However, VLSM of the effect of reward on choices revealed regions of medial and lateral vmPFC, spanning areas 14, 13 and 32 and sgACC, where lesions were associated with reduced sensitivity to reward (Fig. 8). In other words, patients with damage covering these areas scaled up their willingness to work as the reward available increased to a lesser extent. These regions were found in the analysis of all patients, and sgACC damage was also associated with reduced reward sensitivity when limiting the VLSM to patients in the vmPFC group.

In summary, VLSM analyses identified specific regions of vmPFC associated with the decrease in prosociality observed in the group comparisons, and also a distinct region where damage relatively increased willingness to help others. A medial–lateral distinction was found in the areas associated with decreased and relatively increased prosociality, respectively. Regions associated with reduced reward sensitivity covered both the medial wall and more lateral portions of vmPFC, overlapping with the areas linked with changes in prosociality. However, no specific regions were significantly associated with how effort affected choices in our VLSM analysis.

## Discussion

The vmPFC has been linked to social behaviour, effort and reward processing, each crucial for adaptive decision-making[1,12,20,43,46,50,53,54,57,67]. However, previous functional neuroimaging studies cannot reveal causal associations. Lesion studies have used small samples of fewer than ten patients or deployed unincentivized hypothetical choice scenarios for their assessments. Here, in an incentivized decision-making task with real social decisions affecting another person, we show that when another person will benefit, vmPFC lesions lead to earning less money, a greater reluctance to exert effort and application of less force. Patients with vmPFC damage also showed reduced sensitivity to effort but not reward compared with HC participants. Examining areas within the heterogeneous vmPFC revealed specific subregions where damage was associated with reduced reward sensitivity, as well

as a dissociation between medial and lateral areas in which damage to the former led to heightened antisocial behaviours, whereas damage to the latter led to relatively heightened prosocial behaviours. Together, these findings point to multiple causal roles of vmPFC in prosocial behaviour, effort processing and reward-based decision-making, and highlight the importance of distinguishing between distinct subregions for different functions.

The finding that vmPFC lesions overall reduced willingness to help others is consistent with prior work showing a key role of this region in social behaviour[9,13–17,44,46]. However, the results presented here suggest that these prosocial effects are separable from any effects of vmPFC damage on valuing self-benefiting rewards. Previous studies have reported that vmPFC lesions lead to reduced trust and generosity in economic exchanges[14,17]. In those investigations, decisions to reduce benefits for another person also led to increased benefits for oneself, and therefore any differences could reflect altered valuation of self-benefiting rewards. We show that vmPFC damage reduces willingness to benefit another person, with Bayesian evidence supporting no difference in the willingness to benefit oneself. The impact of vmPFC damage on prosocial but not self-benefiting decisions suggests that vmPFC is not processing identity, or self compared with other, but running computations to decide to put in effort into actions that benefit others. Our results also highlight the importance of separating self and other processing experimentally to examine the unique contributions of different brain regions and to highlight their role in processing effort and reward during decision-making for self, other or both.

By leveraging VLSM, we were also able to compare how damage to distinct subregions affected behaviour. Intriguingly, these analyses revealed that damage to medial and lateral vmPFC led to relative differences in prosociality. Damage to area 14 and sgACC on the medial surface led to heightened antisocial behaviour. In contrast, lesions in lateral area 13 enhanced prosocial behaviour relative to damage elsewhere in vmPFC. Outside the literature on social decisions, these different parts of vmPFC have been linked to discrete functions, with broad delineation between how medial and lateral vmPFC portions contribute to decision-making[43,50]. In macaques, area 13 has been associated with encoding the identities of different choice options, which is presumably critical for social decision-making[58,59].

Further support for a distinct causal role comes from studies showing that stimulation and inactivation of area 13 can interfere with value-based decisions[68,69]. In rodents, lateral orbitofrontal cortex has been linked to encoding current goal states[43], whereas medial orbitofrontal cortex is associated with maps of future states[43,54]. Therefore, an alternative idea is that current and future states support differences in antisocial and prosocial behaviours. Future work could compare the different functions subserved by areas 13 and 14 with electrophysiological recordings, to directly compare the properties of neurons in finer-grained medial-versus-lateral vmPFC divisions. It is also important to note that the boundaries of different cytoarchitectonic areas vary considerably between individuals[70], and thus distinctions between these areas may become apparent only with even larger sample sizes. Additionally, separating these areas in samples with heterogeneous damage within vmPFC can be challenging. We do not interpret our results as suggesting a sharp dividing line between these anatomical regions. However, we note that some differentiation of function between lateral and medial vmPFC is consistent with studies across different species and with cytoarchitectonic boundaries that have homologues across species[47,48].

The finding that vmPFC lesions altered sensitivity to effort, without significant differences in reward sensitivity between groups overall, is also notable. Specifically, vmPFC patients were less willing to work when the effort required was minimal than HC participants were. The involvement of vmPFC in processing effort has been reported in human neuroimaging studies[1,3,8,38,71], although not consistently in tasks similar to the one used here[8,23,38]. To our knowledge, no human lesion studies

have evaluated the role of this area in effort-based decision-making. One recent study showed that patients with behavioural variant frontotemporal dementia have disrupted effort sensitivity on hypothetical effort–reward decisions, but real effort-based decisions were not evaluated[72]. Another investigation in humans using position emission tomography reported that dopamine function in vmPFC correlated with willingness to expend effort for reward[71]. Electrophysiological recordings in macaques have also suggested a key role for vmPFC in effort processing, with single neurons tracking willingness to engage in behaviour[73,74]. Here we show, in humans, a causal role for vmPFC in sensitivity to effort, whereby vmPFC lesions lead to lower willingness to engage in low-effort actions.

Considering reward sensitivity, our analysis comparing groups did not find any significant differences in how reward affected choices. As the rewards available increased, all three groups were more motivated to obtain them. This might be considered surprising in the context of extensive functional MRI work demonstrating the encoding of reward in vmPFC[7,75]. However, VLSM analysis did reveal specific subregions where damage was associated with a reduction in how motivation scaled up with increasing available reward. This finding is in line with previous work using neuroimaging[7,75] and lesion methods[20,60,76] that have linked vmPFC to reward processing. The fact that this was limited to specific parts of the vmPFC may help explain the great heterogeneity in previous results[7,54]. Indeed, one study found that damage to different vmPFC subregions had opposing effects on incentivization by reward[60], similar to the opposing effects we observed on prosociality for lateral and medial vmPFC damage. It was notable that lesion mapping correlates of reduced reward sensitivity spanned both the medial areas of vmPFC associated with more antisocial behaviour and the lateral region where damage predicted more prosocial behaviour.

Work on value-based decision-making involving other costs, such as temporal costs in delay discounting tasks, has suggested that vmPFC damage may lead to stronger value discounting[77–79]. However, again evidence is inconsistent between studies[80]. These experiments also typically tested smaller sample sizes, due to the difficulty of accessing patients who have focal brain damage. In the current study, discounting by effort, as captured by a computational parameter ($K$), was higher in patients with vmPFC damage, and this was driven by the prosocial condition. This highlights the importance of computational modelling approaches for distinguishing the different latent contributions to behaviour and cognition between groups. A previous study showed that differences in economic decision-making between vmPFC patients and controls were not related to differences in decision noise/choice consistency[14], which is also what we found here. This computational neurology approach, where focal lesions are related to specific parameters, is important for future work. These different computational parameters could map onto different brain areas, can be used as inputs in VLSM and can be related to transdiagnostic symptoms when studying larger samples, all promising approaches for future research.

In addition to the new insights offered by the current study, there are limitations. While we had a relatively large sample of patients compared with many previous studies, it would be helpful to have an even larger sample for analyses such as lesion–symptom mapping. We were able to identify discrete subregions of vmPFC, but there could be others that would be revealed only in a larger sample. Second, while the experimental task we used had several benefits in independently manipulating the recipient, effort and reward, all aspects of the choice were displayed on each trial. This means that our results cannot evaluate the impact of vmPFC damage on how people learn whether to repeat choices that affect themselves or other people. Several studies have linked vmPFC to learning in different contexts[10], and it would be important to test whether similar alterations are apparent during learning about social behaviour, effort and reward. Finally, although the current study focused on the localization of function, the vmPFC is highly connected to other brain areas that are also crucial for decision-making and

social behaviour, including different portions of the ACC and subcortical areas such as the striatum and amygdala[46]. Future studies might profitably examine how these different regions work in combination.

In conclusion, we show that vmPFC is a crucial brain region for motivation to engage in actions that benefit other people, including choosing to exert effort to help others and energizing the actions that do so. We also highlight the distinct roles of different vmPFC subregions, with damage to medial and lateral areas having dissociable effects on willingness to help others. Sensitivity to effort costs was also reduced by vmPFC lesions overall, whereas damage to specific subregions decreased sensitivity to reward. These findings provide causal evidence of the multiple roles of vmPFC in social behaviours and value-based decision-making, which is critical for understanding the fundamental functions of this important brain area.

## Methods

The research complied with all relevant ethical regulations and the Oxford University Medical Sciences Interdivisional Research Ethics Committee and National Health Service Health Research Authority approved the protocol (Ethics Ref. No. 18/LO/2152). All participants provided written informed consent. The participants were paid at a rate of £10 per hour plus expenses, plus an additional payment of up to £5 depending on the number of credits they earned for themselves during the task. They were also told that the number of credits they earned in the prosocial condition would translate into an additional payment of up to £5 for the other participant. The approved protocol covered this mild deception (see below). The study was not preregistered.

### Participants

We recruited three groups of participants: one with focal damage to vmPFC and one with lesions elsewhere, both from a database of 453 neurological patients, as well as healthy age- and gender-matched controls from university databases and the community. The vmPFC lesion group included 25 patients with vmPFC damage (age range 37–76 years; mean age = 56.44 years; 14 females). Two other patients with vmPFC lesions took part but were excluded from all analyses for not following the task instructions or not understanding the task. The LC group included 15 patients with lesions in areas outside vmPFC (age range 28–74 years; mean age = 56.00 years; 10 females). The HC group included 40 participants with no brain damage (age range 36–67 years; mean age = 60.00 years; 23 females), giving a total sample of $n = 80$ for group-based analyses. No statistical methods were used to predetermine sample sizes, but our sample sizes are larger than those reported in previous publications[13–21]. Finally, five additional patients with lesions affecting more dorsal regions of mPFC or the ACC also completed the task (age range, 49–66 years; mean age, 57.40 years; 2 females). These patients were only included in the lesion mapping analysis (see below), which had a total sample of 45 patients. We did not include patients with dorsal mPFC or ACC lesions in the main between-group analyses, as the damage was not within the vmPFC region of interest but also was not distinct enough to include in the LC group. The results from additional control analyses repeating all the between-group (G)LMMs including these patients in the LC group ($n = 20$) showed the same findings (Supplementary Tables 2–7). Lesion locations were classified from MR imaging or CT scans (see below) by a neurologist (S.G.M.).

Across the vmPFC and LC groups, most lesions were caused by subarachnoid haemorrhage. Three patients' damage was caused by tumour (2 vmPFC and 1 LC) and one LC patient's by head injury. In all cases, the brain damage occurred at least 24 months prior to testing. Of the 45 patients, 15 were hypertensive (10 vmPFC and 5 LC), 4 were on antidepressants (1 vmPFC and 3 LC) and 2 took pregabalin (1 vmPFC and 1 LC). We carefully screened the participants and recruited them such that there were no differences between any pair of groups in terms of gender ($\chi^2_3 = 1.17, P = 0.76$), age (all $P > 0.19$), apathy (Apathy Motivation Index[81]; all $P > 0.14$) or performance on a neuropsychological test of

cognitive abilities and visual attention (Trail Making Test (TMT)[82]; all Part A $P > 0.20$, all Part B $P > 0.37$). The vmPFC patients also did not differ from either control group in self-reported levels of depression (Beck Depression Inventory[83]; all $P > 0.11$) and were matched to LC patients in the extent of their education ($P = 0.40$; Supplementary Table 1).

## Lesion identification

Of the 45 patients, all except two had MR imaging (31 with 2 mm isotropic T1 MRI, 11 with non-isotropic $2 \times 2 \times 5$ mm T1 images and one with only FLAIR images). Two cases had only a CT scan, as they had metal surgical clips and an implantable defibrillator. Each patient's lesion was traced manually onto their brain scan by a neurologist (S.G.M.) using FSL[64] onto the MNI152 template before behavioural testing. The mean lesion volume was 3.04 cm³ (s.d. 2.93 cm³), and the range was 0.02 to 10.11 cm³. Overall lesion volume did not correlate with any of the variables included in the VLSM analysis (recipient effect on choices, recipient effect on discounting parameters, effort effect on choices and reward effect on choices), either across all patients (Supplementary Fig. 3) or within the vmPFC group. An overlap map was constructed by counting in each voxel the number of patients who had a >10% degree of lesion (Fig. 1).

## Procedure

The participants attended a single in-person testing session that started with an assessment with a neurologist (S.G.M.). The experimenters were therefore not blind to group, and groups were naturally occurring rather than based on randomization. Next, we conducted the 'role assignment' procedure (Fig. 2b; see below), and then the participants completed the physical-effort-based decision-making task[24,37] presented in Psychtoolbox (v.3), three other experimental tasks (to be reported elsewhere) and additional questionnaires in Qualtrics. No participants reported that they did not believe their choices affected another person.

**Task.** Physical effort was operationalized as the amount of force participants exerted on a handheld dynamometer. Before the task instructions, the participants were instructed to squeeze the dynamometer as hard as they could. The participants were provided with visual feedback while doing so and encouraged to reach a line that was 110% of their current MVC, on the second and third thresholding trials. After this thresholding procedure, the participants were introduced to another participant anonymously (see 'Role assignment') and then received the task instructions. The participants practised each of the five effort levels twice to ensure that they could be achieved. On each trial of the main task, the participants chose whether to accept a work offer or rest. Rest always earned one credit and required no effort. The work offer presented a variable high-effort, high-reward option of the same duration. The reward available for work varied from two to ten credits (in two-credit increments), and the effort required ranged from 30% to 70% (in 10% increments) of the participant's MVC. The level of effort required for each offer was represented using coloured portions of a pie chart (Fig. 2a). Rewards (credits) on offer for each option were written in colour below. The participants were aware that credits earned corresponded to later compensation but were not made aware of the exchange rate while completing the task. Critically, each trial also varied in whether the reward was for the participant (self condition) or the other person (prosocial condition). The participants had four seconds to choose between the rest and work offers. If they failed to choose an option, they were awarded zero credits after the full trial duration. After choosing, the participants were shown a screen with a yellow horizontal bar on an empty vertical box. The horizontal bar represented the level of effort required; the box was filled according to the force the participants exerted on the dynamometer, providing feedback in real time. For a trial to be considered successful and credits obtained, the participants had to accumulate at least one second at or above the required force level across the three-second force period. Each participant completed 75 interleaved trials per recipient (self or other) broken into three blocks, with a minute break in between each block to rest and prevent the build-up of fatigue.

**Role assignment.** The participants were introduced to another participant who was a confederate, as in previous studies of social decision-making[37] (Fig. 2b). The participants were instructed not to speak and wore a glove to hide any physical characteristics and to ensure they were anonymous to one another. A second experimenter brought the confederate to the other side of the door; the confederate was also instructed not to speak and wore a glove. The participants only ever saw the gloved hand of the confederate, but they waved to each other to make it clear there was another person there. The experimenter tossed a coin to determine who picked a ball from the box first and then told the participants which roles they had been assigned to, on the basis of the ball that they picked. However, the procedure ensured that the participants were always allocated the role of the person performing the effort task, and they were led to believe that the other participant (the confederate) would be performing different tasks. We emphasized that the other participant would be unaware of the task performed by the participant, so any reward given would be anonymous. This procedure used mild deception to minimize as much as possible the role of social preferences or reciprocity[84] in motivating prosocial behaviour.

**Apathy Motivation Index.** Apathetic traits were measured using the Apathy Motivation Index[81]. This index is an 18-item scale used to measure individual differences in apathy and motivation. The participants were asked to indicate their agreement with each item on a five-point Likert scale from 0 to 4. All items are scored such that higher scores indicate higher apathy.

**Beck Depression Inventory.** Depressive symptoms were measured using the 21-item Beck Depression Inventory[83]. The participants rated their agreement on a four-point Likert scale (0–3) for each item, with higher scores indicating higher depression.

**Trail Making Test.** The TMT[82] consists of two parts (A and B) that are requested to be performed as quickly and accurately as possible. TMT-A requires participants to draw lines sequentially connecting, in ascending order, 25 numbers randomly distributed on a sheet of paper (that is, 1–2–3–4 and so on) and is thought to be a measure of visual attention. In TMT-B, the participant must alternate between numbers (1–13) and letters (A–L) while connecting them (that is, 1–A–2–B–3–C and so on); this is thought to be a measure of executive function. The score on each part represents the amount of time required to complete the task.

## Statistical analysis

We analysed the data using R[85] (v.3.6.2) with R Studio[86] (v.1.4.1106). Analysis of behavioural data and computational parameters (see below) used (G)LMMs (the glmer or lmer function in the lme4 package[87] v.1.1-27.1; see Supplementary Methods for the full details of the models). $BF_{01}$ values for non-significant results were calculated using paired and independent Bayesian $t$-tests (ttestBF function, BayesFactor package[88] v.0.9.12-4.4) with the default prior. $BF_{01}$ corresponds to how many times more likely the data are under the null hypothesis of no difference than under the alternative hypothesis that there is a difference. A $BF_{01}$ larger than 3 (equal to $BF_{10}$ less than 1/3) is considered substantial evidence in favour of the null hypothesis, whereas a $BF_{01}$ between 1/3 and 3 indicates that the data cannot differentiate between hypotheses[89]. Simple contrasts between groups used independent parametric ($t$-test) or non-parametric (Wilcoxon two-sided signed rank test) comparisons.

## Computational modelling

We quantified discounting of reward by effort ($K$) and decision consistency ($\beta$) by comparing multiple models that represent different plausible theories of discounting. These included all combinations

of whether the same parameter applied across recipient conditions ($1K$ and $1\beta$) or whether there were two, recipient-specific parameters for discounting ($2K$: $K_{self}$ and $K_{other}$) and decision noise ($2\beta$: $\beta_{self}$ and $\beta_{other}$). We also varied whether the shape of the discount function was linear, hyperbolic or parabolic[32], creating a total of 12 models (Supplementary Methods). The models were fitted to the choice data using an iterative maximum a posteriori (MAP) approach as previously applied[61,62,90], implemented in MATLAB (v.2019b, The MathWorks Inc). See Supplementary Methods for the full details of the MAP approach. All code for model fitting and simulations can be found at https://doi.org/10.17605/osf.io/xdnek (ref. 91). Fitting the data across groups using this method provides the most conservative comparison and is more robust to the influence of outliers than single-step maximum likelihood estimation[92]. It is therefore recommended over single-step methods where it is possible to implement[92].

### Model identifiability and parameter recovery

We used simulated data to establish that the model comparison procedure could correctly choose the best model and that the parameters could be accurately estimated on the basis of our trial schedule[93,94]. For model identifiability, we simulated data for 100 artificial agents on the basis of each of the 12 models, drawing the parameters randomly from a flat distribution between an upper and a lower bound ($0 < K < 1.5$, $0 < \beta < 4$). Simulating ten datasets from each model and fitting each with the same MAP approach and comparison procedure above generated confusion matrices showing the average XP across the ten runs (Fig. 4a). For parameter recovery, we simulated data using a grid of values to cover the full possible ranges of the four parameters ($K_{self}$, $K_{other}$, $\beta_{self}$ and $\beta_{other}$) in the winning model across 900 simulated agents ($K$ values: 0, 0.3, 0.6, 0.9, 1.2 and 1.5; $\beta$ values: 0, 1, 2, 3 and 4, all with added noise drawn from a normal distribution $\times 0.05$). As with model identifiability, we fit the simulated data using the MAP approach applied to data from the participants and created a confusion matrix of the correlations between simulated and fitted parameter values (Fig. 4c).

### Voxel-based Lesion Symptom Mapping

Four behavioural variables of interest were chosen for VLSM on the basis of our specific hypotheses:

1. Effect of recipient (other versus self) on $K$ parameters from the computational model
2. Effect of recipient (other versus self) on choices
3. Effect of effort on choices
4. Effect of reward on choices

The effects of recipient on $K$ parameters and choices examined whether damage to specific subregions of vmPFC was driving the decreased prosociality on these measures observed in the between-group analysis. This analysis also tested whether there were any regions where damage was associated with the reverse effect of increased prosociality, as has been suggested by some previous research[44]. We included the recipient effect on $K$ parameters to capture the latent variable of discounting, in line with our computational neurology approach, and the effect on choices as the main observable behavioural measure on the task. VLSM on the effects of effort and reward on choices considers regions associated with effort and reward sensitivity. For effort, this tests whether the decreased effort sensitivity we observed in vmPFC patients in the group analysis was linked to specific vmPFC subregions. We also included the effect of reward on choices in the VLSM despite no group differences in reward sensitivity, as previous work has strongly linked vmPFC to reward processing. This tests the hypothesis that damage to vmPFC does affect reward sensitivity, but the nature of these effects, only in specific subregions, means this was not captured in the group comparisons.

We quantified these variables for each participant by extracting the participant-level random effect of the relevant variable from adjusted GLMMs of choices and $K$ parameters (ranef function, lme4 package[87]). These GLMMs included data from the five additional patients excluded from the group analysis (see above) and did not model any interactions or group as a predictor, as this would have accounted for some of the individual variance, which is the focus of VLSM. The participant-level random effect values were ranked to remove skew from the distribution of residuals[66] and $z$-scored, as required for the nature of our design, before being put into FSL (v.6.0.6.2)[64] design files. We then used FSL's randomise function to run permutation-based VLSM analysis[65,66], which compares patients with damage at each voxel to all other patients (see https://doi.org/10.17605/osf.io/xdnek (ref. 91) for the analysis code). FSL implements the latest developments in brain-based analyses, while remaining regularly updated and open source. FSL also enables the use of TFCE, which maximizes power and relies on non-arbitrary definitions of cluster size[65] and is not currently available in other lesion-mapping toolboxes (for example, LESYMAP and NiiStat). The patients' lesion maps were mirrored to increase power, as we did not have hypotheses about laterality, creating symmetrical masks. Voxels were included in the VLSM only if at least five patients had damage in that voxel to some extent[60,63] (Supplementary Fig. 4).

We generated $P$ values using permutation-based TFCE in randomise with 5,000 permutations and FSL's default TFCE settings, optimized for this type of data[65,66]. Permutation testing runs the same analysis many times with the data shuffled randomly to calculate voxel-wise $P$ values that capture the probability that the real effect could be due to random noise. It therefore better reflects the nature of the data, relies on fewer assumptions than other methods and can be combined with the benefits of TFCE[66]. We applied a Bonferroni correction for multiple comparisons across the four behavioural regressors ($P < 0.0125$) to the uncorrected maps from the permutation-based TFCE results. For visualization, we applied binarized masks of the significant areas from each analysis to the $t$ values. We also extracted the extent of damage in these regions for each participant to plot against the relevant behavioural variable and separately for self and other average choices or $K$ parameters in the case of recipient effects.

### Reporting summary

Further information on research design is available in the Nature Portfolio Reporting Summary linked to this article.

## Data availability

The data are available at https://doi.org/10.17605/osf.io/xdnek. Source data are provided with this paper.

## Code availability

The code for modelling and analysis is available at https://doi.org/10.17605/osf.io/xdnek.

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

## Acknowledgements

This work was supported by a Medical Research Council Fellowship (MR/P014097/1 and MR/P014097/2), a Christ Church Junior Research Fellowship, a Christ Church Research Centre Grant, a Jacobs Foundation Research Fellowship, a Leverhulme Prize (PLP-2021-196) a Wellcome Trust/Royal Society Sir Henry Dale Fellowship (223264/Z/21/Z) and a UKRI EPSRC Frontiers Research Guarantee/ERC Starting Grant (EP/X020215/1) to P.L.L.; a Biotechnology and Biological

Sciences Research Council David Phillips Fellowship (BB/R010668/1) and Biosciences and Biotechnology Research Council Future Leader Fellowship (BB/M013596/1) awarded to M.A.J.A.; Wellcome Trust Principal Fellowship (098282/Z/12/Z and 206330/Z/17/Z) and National Institute for Health Research (NIHR) Oxford Health Biomedical Research Centre funding to M.H.; and a Clinician Scientist Fellowship (MR/P00878/X) and Leverhulme Research Grant no. 2018-310 to S.G.M. The research was also funded/supported by the NIHR Oxford Biomedical Research Centre. For open access, the authors have applied a CC BY public copyright licence to any Author Accepted Manuscript version arising from this submission. The views expressed are those of the authors and not necessarily those of the NHS, the NIHR or the Department of Health. The funders had no role in study design, data collection and analysis, decision to publish or preparation of the manuscript. We also thank our colleagues who acted as the other participants during the study.

## Author contributions

Conceptualization: P.L.L., M.A.J.A., M.H. and S.G.M. Methodology: P.L.L., D.D., A.A. and S.G.M. Investigation: P.L.L., D.D., A.A. and D.S.J. Formal analysis: P.L.L. and J.C. Writing—original draft: P.L.L. and J.C. Writing—review and editing: P.L.L., J.C., M.A.J.A., M.H. and S.G.M. Funding acquisition: P.L.L. and M.H. Supervision: P.L.L., M.H. and S.G.M.

## Competing interests

The authors declare no competing interests.

## Additional information

**Correspondence and requests for materials** should be addressed to Patricia L. Lockwood or Jo Cutler.

# Reporting Summary

## Statistics

For all statistical analyses, confirm that the following items are present in the figure legend, table legend, main text, or Methods section.

| n/a | Confirmed | |
|---|---|---|
| ☐ | ☒ | The exact sample size ($n$) for each experimental group/condition, given as a discrete number and unit of measurement |
| ☐ | ☒ | A statement on whether measurements were taken from distinct samples or whether the same sample was measured repeatedly |
| ☐ | ☒ | The statistical test(s) used AND whether they are one- or two-sided<br>*Only common tests should be described solely by name; describe more complex techniques in the Methods section.* |
| ☐ | ☒ | A description of all covariates tested |
| ☐ | ☒ | A description of any assumptions or corrections, such as tests of normality and adjustment for multiple comparisons |
| ☐ | ☒ | A full description of the statistical parameters including central tendency (e.g. means) or other basic estimates (e.g. regression coefficient) AND variation (e.g. standard deviation) or associated estimates of uncertainty (e.g. confidence intervals) |
| ☐ | ☒ | For null hypothesis testing, the test statistic (e.g. $F$, $t$, $r$) with confidence intervals, effect sizes, degrees of freedom and $P$ value noted<br>*Give P values as exact values whenever suitable.* |
| ☐ | ☒ | For Bayesian analysis, information on the choice of priors and Markov chain Monte Carlo settings |
| ☐ | ☒ | For hierarchical and complex designs, identification of the appropriate level for tests and full reporting of outcomes |
| ☒ | ☐ | Estimates of effect sizes (e.g. Cohen's $d$, Pearson's $r$), indicating how they were calculated |

*Our web collection on statistics for biologists contains articles on many of the points above.*

## Software and code

Policy information about availability of computer code

| Data collection | Psychtoolbox version 3 and Qualtrics (website, no versions) |
|---|---|
| Data analysis | MATLAB 2019b, R (version 3.6.2) with R studio (version 1.4.1106) and FSL (version 6.0.6.2).<br><br>Code availability<br><br>Code for modelling and analysis is available at https://doi.org/10.17605/osf.io/xdnek |

For manuscripts utilizing custom algorithms or software that are central to the research but not yet described in published literature, software must be made available to editors and reviewers. We strongly encourage code deposition in a community repository (e.g. GitHub). See the Nature Portfolio guidelines for submitting code & software for further information.

## Data

Policy information about <u>availability of data</u>

All manuscripts must include a <u>data availability statement</u>. This statement should provide the following information, where applicable:
- Accession codes, unique identifiers, or web links for publicly available datasets
- A description of any restrictions on data availability
- For clinical datasets or third party data, please ensure that the statement adheres to our <u>policy</u>

> Data are available at https://doi.org/10.17605/osf.io/xdnek

## Research involving human participants, their data, or biological material

Policy information about studies with <u>human participants or human data</u>. See also policy information about <u>sex, gender (identity/presentation), and sexual orientation</u> and <u>race, ethnicity and racism</u>.

| | |
|---|---|
| Reporting on sex and gender | The three groups were matched on self-reported gender: 25 patients with vmPFC damage (mean age=56.44; 14 females), 15 lesion control (LC) patients with damage to areas outside vmPFC (mean age=56.00; 10 females), and 40 healthy control (HC) participants (mean age=60.00; 23 females). No further analysis on sex or gender was required for the research questions and as splitting the samples based on any characteristic would compromise statistical power. |
| Reporting on race, ethnicity, or other socially relevant groupings | These variables were not included as they were not required for the research and as splitting the samples based on any characteristic would compromise statistical power. |
| Population characteristics | Please see behavioural & social sciences study design section |
| Recruitment | We recruited three groups of participants, one with focal damage to vmPFC and one with lesions elsewhere, from a database of 453 neurological patients, as well as healthy age and gender-matched controls from university databases and the community. Self-selection bias of bias due to the requirement of ability to attend in-person experiment using physical effort mean the sample only includes participants with less severe effects of brain lesions on motivation, a more conservative test of group differences than if vmPFC patients with most severe impacts tested. |
| Ethics oversight | The research complied with all relevant ethical regulations and the Oxford University Medical Sciences Inter Divisional Research Ethics Committee and National Health Service Health Research Authority approved the protocol (Ethics Ref. 18/ LO/2152). |

Note that full information on the approval of the study protocol must also be provided in the manuscript.

# Field-specific reporting

Please select the one below that is the best fit for your research. If you are not sure, read the appropriate sections before making your selection.

☐ Life sciences     ☒ Behavioural & social sciences     ☐ Ecological, evolutionary & environmental sciences

For a reference copy of the document with all sections, see <u>nature.com/documents/nr-reporting-summary-flat.pdf</u>

# Behavioural & social sciences study design

All studies must disclose on these points even when the disclosure is negative.

| | |
|---|---|
| Study description | Quantitative experimental with naturally occurring groups (see below). Participants in each of three groups completed a effort-based decision-making task that manipulated effort, reward and recipient (self or other) independently. Dependent variables measured were the number of credits earned, choices to exert effort, force exerted and computational parameters from effort discounting models. |
| Research sample | We compared 25 patients with vmPFC damage (mean age=56.44; 14 females) to two control groups. 15 lesion control (LC) patients had damage to areas outside vmPFC (mean age=56.00; 10 females), and 40 healthy control (HC) participants did not have any brain damage (mean age=60.00; 23 females). Patients across both lesion groups were recruited from a database of 453 neurological patients with lesion control patients and healthy controls selected to age and gender-match the vmPFC patient group. The sample was not recruited to be representative of any specific population but the rationale was to enable us to test the impact of vmPFC lesions specifically, compared to patients with lesions elsewhere and participants without brain damage. |
| Sampling strategy | Convenience sample from neurological database based on lesion location (vmPFC or lesion control) and ability to complete an in-person study using an effort-based decision making task. No sample size calculation was performed. The sample sizes were chosen based on the maximum number of vmPFC and lesion control patients possible, as these patients are extremely rare and a difficult sample to recruit and test. The sample sizes are approximately double the number used in most previous research and the research is also sufficiently powered given the substantial number of trials in the experimental task. |

| | |
|---|---|
| Data collection | Data were collected on a computer. Participants attended a single in-person testing session that started with an assessment with a neurologist (SGM). Experimenters were therefore not blind to group. |
| Timing | March 2017 - August 2019 |
| Data exclusions | Two other patients with vmPFC lesions took part but were excluded from all analyses for not following the task instructions or not understanding the task. Five additional patients with lesions affecting more dorsal regions of mPFC or the anterior cingulate cortex (ACC) also completed the task (age range=49-66, mean=57.40; 2 females). These patients were only included in the lesion mapping analysis (see below), which had a total sample of 45 patients. We did not include patients with dorsal mPFC or ACC lesions in the main between-group analyses, as the damage was not within the vmPFC region of interest, but also was not distinct enough to include in the lesion control group. |
| Non-participation | No participants dropped out. |
| Randomization | Allocation to groups based on lesion location (vmPFC or lesion control) or no lesion (healthy controls). The three groups were carefully matched, with no differences in gender, age, cognitive ability, or levels of apathy. The two lesion groups also did not differ from each other in education or depression (see Methods and Table S1). However, the vmPFC and LC groups did differ from healthy controls in level of education. We therefore repeated all our behavioural analysis controlling for education as a covariate. Controlling for education did not change any of our key results or inferences regarding group differences in prosocial behaviour (see Tables S2-S7). |

# Reporting for specific materials, systems and methods

We require information from authors about some types of materials, experimental systems and methods used in many studies. Here, indicate whether each material, system or method listed is relevant to your study. If you are not sure if a list item applies to your research, read the appropriate section before selecting a response.

## Materials & experimental systems

| n/a | Involved in the study |
|---|---|
| ☒ | Antibodies |
| ☒ | Eukaryotic cell lines |
| ☒ | Palaeontology and archaeology |
| ☒ | Animals and other organisms |
| ☒ | Clinical data |
| ☒ | Dual use research of concern |
| ☒ | Plants |

## Methods

| n/a | Involved in the study |
|---|---|
| ☒ | ChIP-seq |
| ☒ | Flow cytometry |
| ☒ | MRI-based neuroimaging |

## Plants

| | |
|---|---|
| Seed stocks | *Report on the source of all seed stocks or other plant material used. If applicable, state the seed stock centre and catalogue number. If plant specimens were collected from the field, describe the collection location, date and sampling procedures.* |
| Novel plant genotypes | *Describe the methods by which all novel plant genotypes were produced. This includes those generated by transgenic approaches, gene editing, chemical/radiation-based mutagenesis and hybridization. For transgenic lines, describe the transformation method, the number of independent lines analyzed and the generation upon which experiments were performed. For gene-edited lines, describe the editor used, the endogenous sequence targeted for editing, the targeting guide RNA sequence (if applicable) and how the editor was applied.* |
| Authentication | *Describe any authentication procedures for each seed stock used or novel genotype generated. Describe any experiments used to assess the effect of a mutation and, where applicable, how potential secondary effects (e.g. second site T-DNA insertions, mosiacism, off-target gene editing) were examined.* |

