## [Peer Review File · Nature Human Behaviour]

Peer Review Information

Journal: Nature Human Behaviour

Manuscript Title: Human ventromedial prefrontal cortex is necessary for prosocial motivation

Corresponding author name(s): Patricia L. Lockwood and Jo Cutler

Reviewer Comments & Decisions:

Decision Letter, initial version:

18th September 2023

Dear Dr Lockwood,

Thank you once again for your manuscript, entitled "Human ventromedial prefrontal cortex is necessary for prosocial motivation", and for your patience during the peer review process.

Your Article has now been evaluated by 3 referees. You will see from their comments copied below that, although they find your work of potential interest, they have raised quite substantial concerns. In light of these comments, we cannot accept the manuscript for publication, but would be interested in considering a revised version if you are willing and able to fully address reviewer and editorial concerns.

We hope you will find the referees' comments useful as you decide how to proceed. If you wish to submit a substantially revised manuscript, please bear in mind that we will be reluctant to approach the referees again in the absence of major revisions. We are committed to providing a fair and constructive peer-review process. Do not hesitate to contact us if there are specific requests from the reviewers that you believe are technically impossible or unlikely to yield a meaningful outcome.

To guide the scope of the revisions, the editors discuss the referee reports in detail within the team, including with the chief editor, with a view to (1) identifying key priorities that should be addressed in revision and (2) overruling referee requests that are deemed beyond the scope of the current study. We hope that you will find the prioritised set of referee points to be useful when revising your study. Please do not hesitate to get in touch if you would like to discuss these issues further.

1. Reviewer 1 raises important concerns about the VLBM analyses. Specifically, the reviewer is concerned about the seemingly arbitrary chosen criteria for defining significant results, as well as the interpretation of the results. We ask that you use permutation-based thresholding with full correction for multiple comparisons as recommended by Reviewer 1, and revise the interpretation of the results of the VLBM analyses.

2. Reviewer 1 also asks that you demonstrate the robustness of your results by including the five patients with medial PFC/ACC damage in the lesion control group in your analyses, and demonstrate that this does not undermine the main results in the group level comparisons.

3. In addition, please also conduct additional exploratory analyses as requested by Reviewer 3.

4. We ask that you either a public or at least review-only link to your data and code.

Finally, your revised manuscript must comply fully with our editorial policies and formatting requirements. Failure to do so will result in your manuscript being returned to you, which will delay its consideration. To assist you in this process, I have attached a checklist that lists all of our requirements. If you have any questions about any of our policies or formatting, please don't hesitate

to contact me.

If you wish to submit a suitably revised manuscript, we would hope to receive it within 4 months. I would be grateful if you could contact us as soon as possible if you foresee difficulties with meeting this target resubmission date.

- Include a "Response to the editors and reviewers" document detailing, point-by-point, how you addressed each editor and referee comment. If no action was taken to address a point, you must provide a compelling argument. When formatting this document, please respond to each reviewer comment individually, including the full text of the reviewer comment verbatim followed by your response to the individual point. This response will be used by the editors to evaluate your revision and sent back to the reviewers along with the revised manuscript.
- Highlight all changes made to your manuscript or provide us with a version that tracks changes.

[REDACTED]

Thank you for the opportunity to review your work. Please do not hesitate to contact me if you have any questions or would like to discuss the required revisions further.

Sincerely,

[REDACTED]

Reviewer expertise:

Reviewer #1: VLBM analysis ; effort-based decision making tasks

Reviewer #2: affective neuroscience

Reviewer #3: computational modelling ; cognitive neuroscience

REVIEWER COMMENTS:

Reviewer #1:

Remarks to the Author:

In this manuscript, Lockwood et al. describe a study examining the behavior of participants with damage to the vmPFC in a social effort-based decision-task. The task asked participants to choose between resting and receiving 1 credit, or squeezing a dynamometer with varying levels of grip strength to receive 2-10 credits for themselves or another participant. These credits were converted to a monetary bonus at the end of the task. Compared to groups of healthy controls and participants with brain damage sparing the vmPFC, the vmPFC damaged participants were less willing to choose effortful options for the other participant, and exerted less force on trials where they chose the effortful option to earn rewards for the other participant compared to health controls. The authors model behavior in the task comparing different potential discount functions, and also conduct voxel-based lesion behavior mapping to test which damaged areas are most strongly associated with changes in behavior.

The manuscript addresses an interesting question about the involvement of vmPFC in social decision-making. There are a fairly large number of participants in all three groups, giving more

statistical power than is commonly the case for this kind of experimental neuropsychology study. The modeling section also does many due diligence checks, with comparisons between models and tests of model identifiability and parameter recoverability. However, I have several serious methodological concerns with the VLBM analysis. I am also skeptical about choices to exclude participants from some analyses and not others. I have several other clarification questions that may also affect interpretation of the results.

1. Like whole-brain fMRI analyses, VLBM is a massive univariate analysis that is prone to false alarms due to the high number of multiple comparisons. Rather than use standard controls for multiple comparisons that have been regularly applied in the field for years (e.g. permutation-based thresholding), the authors choose two arbitrary criteria for defining significant results. This is a peculiar and statistically unjustified choice that raises doubts about the significance of these results.

2. The authors claim that their VLBM analysis within the vmPFC group shows that damage to a 'lateral' part of vmPFC 'paradoxically' increases prosocial motivation. This conclusion is based on a fallacious interpretation of the VLBM analysis. VLBM statistics are based on a comparison of patients with damage in each voxel to all other patients. Thus, when conducting this analysis in the vmPFC group alone, all comparisons are made between other patients with vmPFC damage. This result thus simply reflects that the vmPFC patients with more lateral damage are more normal than the patients with medial damage (i.e. the flip of the first VLBM analysis). The results do not indicate any increase in prosocial motivation compared to all patients or healthy controls. Indeed, plots of individual participants' data in Fig 3 and Fig 4 do not show any of the vmPFC damaged participants demonstrating excess prosocial behavior compared to the control groups. To test if damage was associated with increased prosociality anywhere, the authors should have conducted this analysis in all the participants with lesion damage.

3. The authors remove 5 participants with damage along the medial wall in group comparisons (what would be 1/4 of the lesion controls), but include these participants in later VLSM analyses. They state that these patients did not have damage within the vmPFC but were not distinct enough to include in the lesion control group. I don't find this to be a compelling reason to exclude these participants. Given that the vmPFC group has damage that extends up the medial wall outside of vmPFC, I would think that these are precisely the best kind of control participants since they have damage in an area that commonly coincides with vmPFC lesions, but do not have damage in the ROI.

4. Were all groups matched in education? Also, please clarify If the comparisons of neuropsychological testing and depression symptoms in the methods refer to comparisons of healthy controls or lesion controls with the vmPFC group. It would help if this information was detailed in a table for greater clarity and transparency.

5. In several plots, the authors plot the data by rank rather than in their raw values (e.g. Figure 6 and 7). This makes it difficult to interpret the values in these results and compare across figures. Please use real values in any revision.

6. The data and code are not available to reviewers. The depository is currently locked and requires a request for access.

7. Why is the comparison of force exerted (Fig 5) only made between the vmPFC and healthy control group? The lesion controls should be included in this comparison.

8. Please clarify if the post-hoc comparisons labeled in the figures with asterisks are $P < 0.05$ after correcting for multiple comparisons.

9. The VLBM maps appear somewhat strangely constrained to the gray matter (e.g. Figure 8). Were the lesion masks in some way masked to exclude any WM voxels?

Some more minor comments:

1. It would help to have a supplementary figure showing which voxels are actually being included in the VLBM analysis.

2. Please include equations for the GLMM analyses in the methods section.
3. There are several references to cytoarchitectonic regions in the paper (e.g. 14m), but I cannot find an explanation or reference for how these areas were defined here. In general, I think it would be more helpful to have peak MNI coordinates for these effects.
4. It would help if the individual participants were connected by lines (e.g Fig 3 a and b) to show the magnitude of the within-participant changes in behavior between the self and other conditions.
5. There seems to be a mistake on line 386: "As participants were generally less willing..."
6. Aside from concerns with the second VLBM analysis, the idea of differentiating between more medial and lateral parts of vmPFC (an area with the word 'medial in the name!) seems peculiar to me. I also am somewhat skeptical that such fine differences can be resolved in a lesion sample like this with large, messy and heterogeneous damage.

Reviewer #2:

Remarks to the Author:

This study involves a rare sample of 25 ventromedial prefrontal cortex (vmPFC) lesion patients as well as non-vmPFC lesion patients and age/gender matched controls. The authors sought to determine the causal contribution of vmPFC to prosocial motivation, effort, and reward during decision making. The authors used neuroimaging, computational modelling, and a sound task design in an interesting and important study. Findings demonstrate vmPFC damage decreases prosociality such that vmPFC lesion patients earned less, were more reluctant to exert effort, and physically exerted less force when another person would benefit compared to the control groups. Reduced prosociality was related to medial damage with a paradoxical increase in prosocial motivation for lateral vmPFC lesions. The authors speculate based on prior literature that choice options for lateral area 13 is important for social decision making, but caution strong conclusions that divide lateral and medial vmPFC with respect to functions in prosocial decision making.

Overall, this is a study with a strong design, unique sample, solid analysis, and worthy of publication in NHB.

Strengths: Task dissociates reward and effort, unlike economic games that conflate reward with financial cost. Large sample of vmPFC lesion patients. Control samples include matched non-lesion and non-vmPFC lesion patients. Comparison of self versus prosocial motivation highlights that vmPFC lesion is not disrupting motivation or reward processes generally but specifically prosocial motivations.

Minor suggestions: The introduction feels a bit disjointed. It might be better organized by moving the second paragraph up first and talking about the effort motivation followed by prosocial behaviour and finally the neural background. With the young adult / older adult discussion it would be helpful to tie this into the brain somehow – is the theory that the age trend is due to vmPFC maturation? There are several theories suggested for vmPFC involvement in decision making, including processing of self versus other which is interesting. It would be nice for this to be brought back up in the discussion to understand what the authors think underlie findings in this study.

Thank you for the opportunity to review this work. I look forward to seeing it in publication.

Reviewer #3:

Remarks to the Author:

In the manuscript entitled 'Human ventromedial prefrontal cortex is necessary for prosocial motivation', the authors propose to dissociate the contribution of vmPFC to prosocial behaviour (decisions causing real beneficial outcomes for another person), effort, and reward with a decision-making task that manipulates these factors independently. The used computational modeling to disentangle the role of these factors. A large group of patients with rare focal vmPFC lesions (n=

25) was compared to patients with lesions elsewhere (n=15), and healthy age and gender-matched controls (n=40). Participants chose either to rest, or to exert effort to gain rewards, for themselves or another person. They found that patients with lesions here earned less, were more reluctant to exert effort, and physically exerted less force when another person would benefit, compared to both control groups. Using voxel-based lesion mapping, they found that medial damage led to antisocial behaviour, while lateral damage was associated with increased prosocial motivation. Patients with vmPFC lesions also showed reduced sensitivity to effort but not reward overall.

The paper presents a nice design which enables to test participants on social and non-social versions of the same task. The analyses are nice and globally convincing. I nevertheless raise a number of points where clarifications or improvements could be necessary.

At the end of the introduction (line=110) it is stated that 'vmPFC patients earned less', while in the results (lines 153-155) it is said that they 'earned the same amount as lesion controls for themselves' and 'did not significantly differ from healthy controls'. This seems contradictory. Is it because line 110 speaks about the average earning (for self and other)? If yes, I think this is confusing and induces the reader to think that vmPFC patients have an impaired performance even in the self condition. Alternatively, I think I understood later on that they earned less 'when another person would benefit', as written at the end of the sentence. I think the sentence could become clearer if it started with 'When another person would benefit, vmPFC patients ...'. Same situation in the Discussion section, line 458: 'that lesions to vmPFC lead to earning less money' -> 'that, when another person will benefit, lesions to vmPFC ...'. (These are just suggestions; The authors might disagree).

I don't understand how vmPFC patients could earn 'the same amount as lesion controls for themselves (ratio=1.00, ...' while Figure 3a shows a difference in earned credits for self between vmPFC patients (approx. 380) and lesion controls (approx. 390). Could the authors please explain?

Figures 5 and S2 should also include lesion controls. This would help understand why 2 patients from the lesion control group earned less than 200 credits for other (Figure 3a) while accepting to work more for other more than 50% of the time (Figure 3b).

It would be great to add a black vertical line in Figure 7b and 7c (as well as in Figure S5a and S5b) to show where the healthy controls are. In terms of K difference, I take from Figure 6c that they would be around 18-19, thus probably non significantly different from patients with 0-rank damage to Area 13a/b. However, in terms of choice difference, I take from Figure 6b that they would be around 27-28, thus far away from patients with 0-rank damage to area 13a/b. Is it because these patients have a lesion elsewhere? If yes, where?

In the Methods section, lines 713-715, the authors should cite a reference in support of the following claim: 'Fitting the data across groups using this method provides the most conservative comparison and is more robust to the influence of outliers than single-step maximum likelihood estimation (MLE).' Moreover, can't such an iterative, hierarchical model fitting method constraint model parameters for different participants to be drawn from the same higher-level distribution? In other words, doesn't this bias outliers towards the mean of the distribution, and doesn't it pull non-outliers towards outliers?

Didn't the fact that the role of the 'other participant' of the study was played by the authors' colleagues, and that they didn't do the task, constitute a case of deception for the real participants of the task? Were the latter informed afterwards? How did they react? From an ethical point of view, how was deception addressed if it was deception?

There is a mistake in Equation (a) Parabolic in the supplementary material: '(1' is missing after R(t).

Since Figure 3b shows a couple of vmPFC patients which accept nearly 100% of high work options for other, could the authors investigate why they didn't find at least a couple values of beta parameter above 3 for vmPFC patients in the Other condition (like for healthy controls and lesion controls) (Figure S1)? Moreover, could the authors check whether participants around 50% accept in Figure 3b (no matter the group nor the condition) are associated with near-0 values of beta parameter in Figure S1? If not, could they check which participants get near-0 betas and why?

Finally, it would be nice to cite people who have also tested parabolic (vs. linear vs. hyperbolic) discounting of reward by physical effort: Hartmann, M. N., Hager, O. M., Tobler, P. N., & Kaiser, S. (2013). Parabolic discounting of monetary rewards by physical effort. *Behavioural processes*, 100, 192-196.

Author Rebuttal to Initial comments

Reviewer #1:

Remarks to the Author:

In this manuscript, Lockwood et al. describe a study examining the behavior of participants with damage to the vmPFC in a social effort-based decision-task. The task asked participants to choose between resting and receiving 1 credit, or squeezing a dynamometer with varying levels of grip strength to receive 2-10 credits for themselves or another participant. These credits were converted to a monetary bonus at the end of the task. Compared to groups of healthy controls and participants with brain damage sparing the vmPFC, the vmPFC damaged participants were less willing to choose effortful options for the other participant, and exerted less force on trials where they chose the effortful option to earn rewards for the other participant compared to health controls. The authors model behavior in the task comparing different potential discount functions, and also conduct voxel-based lesion behavior mapping to test which damaged areas are most strongly associated with changes in behavior.

The manuscript addresses an interesting question about the involvement of vmPFC in social decision-making. There are a fairly large number of participants in all three groups, giving more statistical power than is commonly the case for this kind of experimental neuropsychology study. The modeling section also does many due diligence checks, with comparisons between models and tests of model identifiability and parameter recoverability. However, I have several serious methodological concerns with the VLBM analysis. I am also skeptical about choices to exclude participants from some analyses and not others. I have several other clarification questions that may also affect interpretation of the results.

Response: Thank you so much for your positive feedback about our work. We are delighted that you found our manuscript addressed an interesting question and appreciated our sample size and due diligence checks on identifiability and model recovery. We are grateful for your time reviewing our work and your valuable comments, which have helped us to significantly improve our manuscript.

1. Like whole-brain fMRI analyses, VLBM is a massive univariate analysis that is prone to false alarms due to the high number of multiple comparisons. Rather than use standard controls for multiple comparisons that have been regularly applied in the field for years (e.g. permutation-based thresholding), the authors choose two arbitrary criteria for defining significant results. This is a peculiar and statistically unjustified choice that raises doubts about the significance of these results.

Response: Thank you for your query regarding our VLSM analysis. We apologise for not providing more details and justification of our analytic approach. Our understanding of the field is that there is a myriad of different approaches, and criteria applied within each, that aim to balance between type I and type II errors (e.g. de Haan & Karnath, 2018; Karnath, Sperber, & Rorden, 2018; Mirman et al., 2018; Sperber & Karnath, 2018) as is the case in fMRI (e.g. Noble, Scheinost, & Constable, 2020). In our original submission, we selected a commonly used threshold in fMRI of $z > 2.3$ to capture where within vmPFC specifically may drive our group-level effects. Setting a minimum cluster size is also common practice in both fMRI and lesion-based analysis and it is recommended that the minimum number of voxels is chosen based on the

number of patients and typical size of the lesions (Karnath et al., 2018). This naturally means the minimum cluster size used varies between studies and we originally chose to report clusters of >10 voxels based on our sample and lesion size.

While these principles mean our criteria were not arbitrary, we agree that our analysis would be even more robust if we were to show similar results using permutation testing and a threshold-free cluster approach. We have now run these analyses which show almost identical results to our original approach. We have updated our manuscript to report the results of these new analyses throughout:

Methods: “We quantified these variables for each participant by extracting the subject-level random effect of the relevant variable from adjusted GLMMs of choices and K parameters. These GLMMs included data from the five additional patients excluded from the group analysis (see above) and did not model any interactions or group as a predictor, as this would have accounted for some of the individual variance, which is the focus of VLSM. We used the randomise function from the FMRIB software library (FSL)⁶⁴ to run permutation-based VLSM analysis^{65,66}. Patients’ lesion maps were mirrored to increase power as we did not have hypotheses about laterality, creating symmetrical masks. Voxels were only included in the VLSM if at least five patients had damage in that voxel to some extent^{45,60}. Each regressor was ranked and then demeaned, as required in FSL for the nature of our design. We generated p values using permutation-based threshold-free cluster enhancement (TFCE) in randomise with 5000 permutations and FSL’s default TFCE settings, optimised for this type of data^{65,66}. Permutation testing runs the same analysis many times with the data shuffled randomly to calculate voxel-wise p values that capture the probability that the real effect could be due to random noise. It therefore better reflects the nature of the data, relies on fewer assumptions than other methods, and can be combined with the benefits of TFCE⁶⁵.

We applied a Bonferroni correction for multiple comparisons across the four behavioural regressors ($p < 0.0125$) to the uncorrected maps from the permutation-based TFCE results^{65,66}. For visualisation, we applied binarized masks of the significant areas from each analysis to the t values. We also extracted the extent of damage in these regions for each participant to plot against the relevant behavioural variable and separately for self and other average choices or K parameters in the case of recipient effects.”

Results: “We only included voxels where at least five patients had damage⁶⁰. We used the FMRIB software library (FSL)⁶¹ to run a permutation-based VLSM with threshold-free cluster enhancement^{65,66} and report significance at $p < 0.0125$ ($p < 0.05$ Bonferroni-corrected across the four behavioural regressors; see Methods).”

There were no areas where damage was significantly associated with the effect of effort on choices at our thresholding criteria (permutation-based TFCE $p < 0.0125$).

Figure 6. Damage to medial portions of vmPFC specifically is associated with reduced prosocial behaviour. (a) Permutation-based, whole-brain, non-parametric voxel-based lesion-symptom mapping (VLSM) reveals subregions of vmPFC on the medial surface where damage is associated with reduced prosociality. This effect was found for both the self-other difference in choices to accept the work over the rest option (orange) and the self-other difference in devaluing rewards by effort (K parameters; red). Specifically, the recipient effect on choices was associated with damage to area 14 (peak 0, 30, -22; see Figure S5) and area 25 (subgenual anterior cingulate cortex, sgACC; peak ± 2 , 16, -6). The recipient effect on discounting K parameters was also associated with damage to area 14 (peak ± 2 , 14, -20) and area 25 (peak ± 2 , 14, -8). Plotting the (ranked) effect of the recipient on choices **(b)** and K parameters

(c) separately for participants with damage or no damage in the areas identified by the relevant VLSM analysis reveals damage decreases prosociality. To further interpret these differences, we also plotted choices and K parameters for self and other separately by damage to the medial vmPFC regions (see Figure S6).

Figure 7. Lateral vmPFC damage is associated with relatively increased prosocial behaviour within the vmPFC patient group. (a) Permutation-based, whole-brain, non-parametric voxel-based lesion-symptom mapping (VLSM) shows damage to a lateral portion of vmPFC is associated with relatively increased prosociality (see Figure S7 for comparison with average behaviour of healthy controls). An overlapping region in area 13 was identified in both the VLSM of the self-other difference in choices (dark blue; peak $\pm 12, 20, -22$) and how rewards were devalued by effort (K parameters; teal;

$\pm 14, 20, -12$; see Figure S5). Plotting the (ranked) self-other difference by whether participants had damage in the identified region shows the relatively increased prosociality in both choices (b) and K parameters (c), compared to patients with damage elsewhere in vmPFC. To further interpret these differences, we also plotted K_{self} and K_{other} separately (see Figure S7).

Figure 8. Damage to medial and lateral portions of vmPFC decreases sensitivity to reward.

Permutation-based, whole-brain, non-parametric voxel-based lesion-symptom mapping (VLSM) shows portions of vmPFC spanning medial and more lateral areas where damage is associated with reduced reward sensitivity. More damage in these subregions, areas 14, 13, 32 and 25, predicted a less positive effect of reward on choices in both the analysis of all patients (light brown; peak 6, 26, - 2) and the vmPFC group only (dark brown; peak $\pm 6, 24, -2$).

2. The authors claim that their VLBM analysis within the vmPFC group shows that damage to a 'lateral' part of vmPFC 'paradoxically' increases prosocial motivation. This conclusion is based on a fallacious interpretation of the VLBM analysis. VLBM statistics are based on a comparison of patients with damage in each voxel to all other patients. Thus, when conducting this analysis in the vmPFC group alone, all comparisons are made between other patients with vmPFC damage. This result thus simply reflects that the vmPFC patients with more lateral damage are more normal than the patients with medial damage (i.e. the flip of the first VLBM analysis). The results do not indicate any increase in prosocial motivation compared to all patients or healthy controls. Indeed, plots of individual participants' data in Fig 3 and Fig 4 do not show any of the vmPFC damaged participants demonstrating excess prosocial behavior compared to the control groups. To test if damage was associated with increased prosociality anywhere, the authors should have conducted this analysis in all the participants with lesion damage.

Response. Thank you for the opportunity to clarify. We did indeed conduct this analysis in all patients with lesion damage, as suggested. For each behavioural regressor, we examined effects in (a) all patients with lesion damage and (b) vmPFC patients only. As you highlight, the interpretation of these analyses differs, and we included them both precisely due to the fact they address separate questions and hypotheses.

Regarding the comment that no lesion patient exhibited more prosocial behaviour than healthy controls, we have also now added a line representing the healthy control mean on the figures used to interpret our lesion mapping results (as suggested by Reviewer 3 and please also see below response to your point 5). This revealed that there are indeed some vmPFC patients who score higher than the average for healthy controls. Nevertheless, we agree that our original phrasing when summarising these findings was not clear enough and describing this result as 'relatively more prosocial' compared to vmPFC patients with damage elsewhere would be more accurate. We have now therefore updated our language throughout the manuscript to ensure that it is clear that higher prosociality is relative to other patients in the vmPFC

group, and not healthy controls. We have also removed the word *paradoxically* from our abstract and throughout to provide a more nuanced interpretation:

Introduction: However, this analysis also identified a more lateral vmPFC subregion where damage was associated with *relatively* increased prosociality, with rewards for others devalued less relative to rewards for self.

Results: Strikingly, this revealed a distinct, lateral portion of vmPFC, putatively in area 13⁴⁹, (**Figure S5**) where damage was associated with *relatively increased* prosociality, compared with damage to other vmPFC subregions, in both choices and effort discounting (K parameters; **Figure 7**). To interpret these findings, we again plotted the K parameters for each recipient separately by whether participants had damage in this vmPFC subregion (**Figure S7**). This showed damage to area 13 was associated with lower values of K_{other} , and therefore *relatively* greater willingness to exert effort for prosocial rewards, but relatively higher values of K_{self} , meaning devaluing self-benefitting rewards to a greater extent when effort was required.

Figure 7. Lateral vmPFC damage is associated with relatively increased prosocial behaviour within the vmPFC patient group. (a) Permutation-based, whole-brain, non-parametric voxel-based lesion-symptom mapping (VLSM) shows damage to a lateral portion of vmPFC is associated with *relatively* increased prosociality (see **Figure S7** for comparison with average behaviour of healthy controls). An overlapping region in area 13 was identified in both the VLSM of the self-other difference in choices (dark blue; peak $\pm 12, 20, -22$) and how rewards were devalued by effort (K parameters; teal; $\pm 14, 20, -12$; see **Figure S5**). Plotting the (ranked) self-other difference by whether participants had damage in the identified region shows the relatively increased prosociality in both choices (b) and K parameters (c), compared to patients

with damage elsewhere in vmPFC. To further interpret these differences, we also plotted K_{self} and K_{other} separately (see Figure S7).

In summary, VLSM analyses identified specific regions of vmPFC associated with a decrease in prosociality observed in the group comparisons, and also a distinct region where damage relatively increased willingness to help others. A medial-lateral distinction was found in the areas associated with decreased and relatively increased prosociality respectively.

Discussion: Examining areas within the heterogeneous vmPFC revealed specific subregions where damage was associated with reduced reward sensitivity, as well as a dissociation between medial and lateral areas that led to heightened antisocial and, in contrast, relatively heightened prosocial behaviours respectively.

Intriguingly, these analyses revealed that damage to medial and lateral vmPFC led to relative differences in prosociality. Damage to area 14 and sgACC on the medial surface, led to heightened antisocial behaviour. In contrast, lesions in lateral area 13 enhanced prosocial behaviour relative to damage elsewhere in vmPFC.

Abstract: Whilst medial damage led to antisocial behaviour, more lateral damage was associated with relatively increased prosocial motivation.

Figure S7. Damage to lateral vmPFC compared to other vmPFC subregions leads to relative increases in willingness to help others. Our voxel-based lesion symptom mapping (VLSM) analysis identified a lateral portion of vmPFC in area 13 where damage was, in contrast, associated with relatively increased prosociality. To further interpret this finding, we extracted the extent of damage for each participant in the region identified in the VLSM analysis of the recipient effect (other vs. self) on K parameters (see Figure 7A). The corresponding effect on choices was associated with a smaller, overlapping region of area 13. Plotting damage in this region against discounting K parameter for other **(a)** and discounting K parameter for self

(b) separately shows the relative increase in (relative prosociality is driven by both lower discounting (higher willingness to work) for other and higher discounting (lower willingness to work) to some extent for self. The dotted line represents the mean value for the healthy control group for comparison.

3. The authors remove 5 participants with damage along the medial wall in group comparisons (what would be 1/4 of the lesion controls), but include these participants in later VLSM analyses. They state that these patients did not have damage within the vmPFC but were not distinct enough to include in the lesion control group. I don't find this to be a compelling reason to exclude these participants. Given that the vmPFC group has damage that extends up the medial wall outside of vmPFC, I would think that these are precisely the best kind of control participants since they have damage in an area that commonly coincides with vmPFC lesions, but do not have damage in the ROI.

Response. *Thank you for this suggestion. We decided a priori, and as part of our recruitment strategy, to include in our vmPFC group patients with damage only to the vmPFC (to capture a group with focal vmPFC damage), and our lesion controls to not include damage on the medial wall outside of vmPFC. As we only had a couple of patients with damage bordering vmPFC, we felt this would not provide enough power to conclude about the effects of vmPFC damage vs. elsewhere. However, we do agree that it is important to show that our group results would be robust to including these additional patients as lesion controls. We have now run these analyses and show that the inclusion of these patients in the lesion control group does not change any of our results or conclusions. We have added the results of these new analyses as an*

additional column in all original supplementary tables (S2-S7, S1 is a new demographic summary please see below). We have also included additional rationale for our approach in the main manuscript:

Methods: Finally, five additional patients with lesions affecting more dorsal regions of mPFC or the anterior cingulate cortex (ACC) also completed the task (age range=49-66, mean=57.40; 2 females). These patients were only included in the lesion mapping analysis (see below), which had a total sample of 45 patients. We did not include patients with dorsal mPFC or ACC lesions in the main between-group analyses, as the damage was not within the vmPFC region of interest, but also was not distinct enough to include in the lesion control group. Results from additional control analyses repeating all the between-group (G)LMMs including these patients in the LC group (n=20) showed the same findings (see column LC n=20 in Tables S2-S7).

Parameter	b	SE	CI low	CI high	t	p	LC n=20
(Intercept)	299.50	21.23	260.36	344.53	80.44	<0.001	<0.001
Recipient (Self	1.25	0.04	1.17	1.33	6.59	<0.001	<0.001
Group (vmPFC vs. HC)	1.20	0.11	1.01	1.44	2.03	0.044	0.100
Group (vmPFC vs. LC)	1.15	0.13	0.91	1.44	1.19	0.23	0.77
Recipient * Group	0.88	0.04	0.81	0.95	-3.07	0.003	0.001

Table S2. Generalised linear mixed-effects model predicting credits

vs. Other)

(vmPFC vs. HC) Recipient * Group 0.87 0.05 0.78 0.97 -2.60
0.010 <0.001

(vmPFC vs. LC)

Note. HC: Healthy controls, LC: Lesion controls. LC n=20: supplementary analysis including the five additional patients with damage on the medial wall, but not focal to vmPFC, in the lesion control group (n=20). Results did not change compared to without these patients.

Table S3. Generalised linear mixed-effects model predicting choices

Parameter	OR	SE	CI low	CI high	z	p	LC n=20
(Intercept)	14.25	6.10	6.16	32.95	6.21	<0.001	<0.001
Effort	0.40	0.08	0.27	0.60	-4.43	<0.001	<0.001
Reward	6.91	1.36	4.70	10.15	9.83	<0.001	<0.001
Recipient (Self vs. Other)	5.73	1.22	3.77	8.70	8.18	<0.001	<0.001
Group (vmPFC vs. HC)	1.83	0.90	0.70	4.82	1.23	0.22	0.21
Group (vmPFC vs. LC)	2.09	1.32	0.60	7.21	1.16	0.25	0.35
Effort * Group (vmPFC vs. HC)	0.44	0.10	0.28	0.69	-3.57	<0.001	<0.001
Effort * Group (vmPFC vs. LC)	0.73	0.21	0.42	1.27	-1.11	0.27	0.40
Reward * Recipient (Self vs. Other)	1.71	0.14	1.46	2.01	6.56	<0.001	<0.001
Recipient * Group (vmPFC vs. HC)	0.60	0.14	0.38	0.94	-2.22	0.026	0.022
Recipient * Group (vmPFC vs. LC)	0.48	0.14	0.27	0.84	-2.57	0.010	0.003
Effort * Reward	0.93	0.06	0.82	1.06	-1.06	0.29	0.27

Note. HC: Healthy controls, LC: Lesion controls. LC n=20: supplementary analysis including the five additional patients with damage on the medial wall, but not focal to vmPFC, in the lesion control group (n=20). Results did not change compared to without these patients.

Parameter	b	SE	CI low	CI high	t	p	LC n=20
(Intercept)	0.08	0.02	0.05	0.12	-13.18	<0.001	<0.001
Recipient (Self vs. Other)	0.54	0.04	0.46	0.64	-7.71	<0.001	<0.001
Group (vmPFC vs. HC)	0.84	0.21	0.52	1.36	-0.72	0.47	0.48
Group (vmPFC vs. LC)	0.67	0.21	0.36	1.24	-1.28	0.20	0.30
Recipient * Group	1.36	0.13	1.12	1.65	3.17	0.002	0.002
Recipient * Group	1.46	0.18	1.15	1.85	3.13	0.002	<0.001

Table S4. Generalised linear mixed-effects model predicting K parameter

vs. Other)

(vmPFC vs. HC)

(vmPFC vs. LC)

Note. HC: Healthy controls, LC: Lesion controls. LC n=20: supplementary analysis including the five additional patients with damage on the medial wall, but not focal to vmPFC, in the lesion control group (n=20). Results did not change compared to without these patients.

Parameter	b	SE	CI low	CI high	t	p	LC n=20
(Intercept)	0.83	0.13	0.61	1.12	-1.24	0.22	0.27
Recipient (Self vs. Other)	1.69	0.16	1.40	2.05	5.41	<0.001	<0.001
Group (vmPFC vs. HC)	1.36	0.27	0.92	2.01	1.56	0.12	0.12
Group (vmPFC vs. LC)	1.22	0.31	0.74	2.02	0.80	0.42	0.59

Recipient * Group	0.85	0.10	0.66	1.08	-1.35	0.18	0.18
Recipient * Group	0.84	0.13	0.61	1.15	-1.12	0.27	0.056

Table S5. *Generalised linear mixed-effects model predicting β parameter*

vs. Other)

(vmPFC vs. HC)

(vmPFC vs. LC)

Note. HC: Healthy controls, LC: Lesion controls. LC n=20: supplementary analysis including the five additional patients with damage on the medial wall, but not focal to vmPFC, in the lesion control group (n=20). Results did not change compared to without these patients.

Parameter	b	SE	CI low	CI high	t	p	LC n=20
(Intercept)	-0.08	0.07	-0.21	0.05	-1.19	0.23	0.36
Effort	0.78	0.04	0.70	0.86	18.26	<0.001	<0.001
Recipient (Self	0.13	0.03	0.06	0.20	3.62	<0.001	<0.001
Reward	0.04	0.01	0.01	0.06	3.16	0.002	0.002
Group (vmPFC vs. HC)	0.10	0.09	-0.07	0.27	1.20	0.23	0.25
Group (vmPFC vs. LC)	0.12	0.11	-0.10	0.33	1.05	0.29	0.93
Effort * Group (vmPFC	0.07	0.05	-0.04	0.17	1.21	0.23	0.23
Effort * Group (vmPFC	-0.13	0.07	-0.27	0.00	-1.94	0.053	0.088
Effort * Recipient (Self	0.03	0.01	0.01	0.05	2.89	0.004	0.004
Recipient * Group	-0.09	0.04	-0.18	-0.01	-2.10	0.036	0.032
Recipient * Group	-0.05	0.06	-0.16	0.05	-0.97	0.33	0.22
Reward * Group	0.00	0.01	-0.03	0.03	-0.18	0.86	0.84
Reward * Group	0.03	0.02	0.00	0.07	1.73	0.084	0.066
Effort * Reward	0.01	0.01	-0.01	0.03	1.03	0.30	0.30
Recipient * Reward	-0.01	0.01	-0.03	0.02	-0.53	0.59	0.54
Effort * Reward *	-0.03	0.01	-0.05	0.00	-2.05	0.041	0.042
Effort * Reward *	-0.02	0.02	-0.05	0.01	-1.51	0.13	0.18
Effort * Recipient *	-0.03	0.01	-0.05	0.00	-2.09	0.037	0.041
Effort * Recipient *	0.00	0.02	-0.03	0.03	0.24	0.81	0.83
Effort * Recipient *	-0.04	0.01	-0.05	-0.02	-3.57	<0.001	<0.001
Recipient * Reward *	0.02	0.02	-0.01	0.05	1.55	0.12	0.11
Recipient * Reward *	0.01	0.02	-0.03	0.04	0.34	0.73	0.66
Effort * Recipient *	0.05	0.01	0.03	0.08	4.21	<0.001	<0.001
Reward * Group (vmPFC vs. HC)	0.05	0.01	0.03	0.08	4.21	<0.001	<0.001
Effort * Recipient * Reward * Group	0.02	0.02	-0.01	0.05	1.38	0.17	0.28

Table S6. Linear mixed-effects model predicting force

vs. Other)

vs. HC)

vs. LC)

vs. Other)

(vmPFC vs. HC)

(vmPFC vs. LC)

(vmPFC vs. HC)

(vmPFC vs. LC)

Group (vmPFC vs. HC)

Group (vmPFC vs. LC)

Group (vmPFC vs. HC)

Group (vmPFC vs. LC)

Reward

Group (vmPFC vs. HC)

Group (vmPFC vs. LC)

(vmPFC vs. LC)

Note. HC: Healthy controls, LC: Lesion controls. LC n=20: supplementary analysis including the five additional patients with damage on the medial wall, but not focal to vmPFC, in the lesion control group (n=20). Results did not change compared to without these patients.

Parameter	b	SE	CI low	CI high	t	p	LC n=20
(Intercept)	-0.30	0.16	-0.62	0.02	-1.86	0.064	0.32
Recipient (Self vs. Other)	0.35	0.11	0.13	0.57	3.17	0.002	0.001
Group (vmPFC vs. HC)	0.54	0.21	0.14	0.95	2.63	0.009	0.059
Group (vmPFC vs. LC)	0.12	0.26	-0.40	0.64	0.46	0.65	0.63
Recipient * Group	-0.32	0.14	-0.59	-0.04	-2.30	0.023	0.020
Recipient * Group	-0.18	0.18	-0.53	0.17	-1.03	0.30	0.13

Table S7. Linear mixed-effects model predicting success

(vmPFC vs. HC)
(vmPFC vs. LC)

Note. HC: Healthy controls, LC: Lesion controls. LC n=20: supplementary analysis including the five additional patients with damage on the medial wall, but not focal to vmPFC, in the lesion control group (n=20). Results did not change compared to without these patients.

4. Were all groups matched in education? Also, please clarify If the comparisons of neuropsychological testing and depression symptoms in the methods refer to comparisons of healthy controls or lesion controls with the vmPFC group. It would help if this information was detailed in a table for greater clarity and transparency.

Response: Thank you for the suggestion to include this information in a table so that it is easily accessible for readers. We have now added this to the supplement (Table S1, reproduced below). The two lesion groups were recruited to be matched in education but had lower levels of education than healthy controls. They also did not differ from each other in depression symptoms, although lesion control patients were more depressed than healthy people. All three groups were matched on a non-social executive function trail-making task.

Results: The three groups were carefully matched, with no differences in gender, age, cognitive ability, or levels of apathy. The two lesion groups also did not differ from each other in education or depression (see Methods and Table S1).

Methods: We carefully screened participants and recruited them such that there were no differences between any pair of groups in terms of gender ($\chi^2_{(3)} = 1.17, p=0.76$), age ($ps > 0.19$), apathy (Apathy Motivation Index⁸¹; $ps > 0.14$), or performance on a neuropsychological test of cognitive abilities and visual attention (Trail Making Test⁸²; Part A $ps > 0.20$, Part B $ps > 0.37$). The vmPFC patients also did not differ from either control group in self-reported levels of depression (Beck Depression Inventory⁸³; $ps > 0.11$) and were matched to lesion controls in the extent of their education ($p=0.40$; Table S1).

Table S1. Summary of demographic variables for each group and between-group comparisons

Variable	HC mean [SD]	vmPFC mean [SD]	LC mean [SD]	HC – vmPFC	LC – vmPFC	HC – LC
Age	60.00 [8.11]	56.44 [10.95]	56.00 [11.81]	0.19	0.99	0.20
Gender	42% M	44% M	33% M	1.00	0.74	0.76
Edu	4.08 [1.14]	2.84 [1.11]	3.27 [1.33]	<0.001	0.40	0.034

AMI	1.19 [0.50]	1.41 [0.57]	1.32 [0.60]	0.14	0.99	0.30
BDI	7.75 [9.34]	10.52 [8.27]	14.27 [11.54]	0.11	0.36	0.044
Trail A	23.37 [8.16]	26.28 [9.50]	29.24 [13.64]	0.20	0.65	0.16
Trail B	57.17 [29.76]	66.90 [41.15]	75.32 [37.25]	0.40	0.37	0.081

Note. HC: Healthy controls, LC: Lesion controls, SD: standard deviation of the mean, Edu: Education scale (1-6); AMI: Apathy Motivation Index, BDI: Beck Depression Inventory; Trial A / B: Trail Making Test Part / Part B, M: male; *p* values for comparisons between groups from Wilcoxon two-sided signed rank tests, except gender which used χ^2 tests for proportions.

5. In several plots, the authors plot the data by rank rather than in their raw values (e.g. Figure 6 and 7). This makes it difficult to interpret the values in these results and compare across figures. Please use real values in any revision.

Response: Thank you for the suggestion to include non-ranked values. We used ranked values because the VLSM is non-parametric. Ranked values capture the raw input to the VLSM, and therefore ensure that the plots reflect the statistical analysis used. The raw versions of these values for the VLSM and Figure 6 & Figure 7 are modelled participant-level random effects of each variable, and are therefore in themselves not simple to interpret. However, we created Figure S6 & Figure S7 to aid interpretation and agree that changing to raw values would improve these figures. We have now updated Figure S6A-D and Figure S7A-B to use raw values. As these plots show choices and K values for self and other separately, the raw values are much more comparable to the behavioural results in Figure 3 and 4 and therefore easier to interpret. We have also made all of the lesion mapping plots compare damage to no damage to help comparison with the line for healthy controls (added in response to Reviewer 3 and your point 2). This also means the extent of damage is now never plotted as ranked values.

Figure S6. Damage to medial vmPFC regions decreases willingness to exert effort for others and increases willingness to obtain self-benefitting rewards. To further interpret the voxel-based lesion symptom mapping (VLSM) finding of decreased prosociality with damage to portions of vmPFC on the medial wall (areas 14 and 25), we plotted each recipient separately to examine how damage affects choices for other **(a)** choices for self **(b)** discounting K parameters for other **(c)** and discounting K parameters for self **(d)**. We categorised participants by whether they had damage in the areas identified in the VLSM analysis for each effect (choices and K ; see Figure 6A). **Patients with no damage to the identified region had damage elsewhere in the brain.** These plots suggest that decreased prosociality shown by the recipient effects (other vs. self) in the VLSM are driven by both lower willingness to help the other person and higher willingness to work for oneself in patients with damage to these regions. **The dotted line represents the mean value for the healthy control group for comparison.**

Figure S7. Damage to lateral vmPFC compared to other vmPFC subregions leads to relative increases in willingness to help others. Our voxel-based lesion symptom mapping (VLSM) analysis identified a lateral portion of vmPFC in area 13 where damage was, in contrast, associated with relatively increased prosociality. To further interpret this finding, we extracted the extent of damage for each participant in the region identified in the VLSM analysis of the recipient effect (other vs. self) on K parameters (see Figure 7A). The corresponding effect on choices was associated with a smaller, overlapping region of area 13. Plotting damage in this region against discounting K parameter for other

(a) and discounting K parameter for self (b) separately shows the relative increase in relative prosociality is driven by both lower discounting (higher willingness to work) for other and higher discounting (lower willingness to work) to some extent for self. The dotted line represents the mean value for the healthy control group for comparison.

We have additionally re-produced the plots from Figure 6 & Figure 7 with the non-ranked values (see below) which show the pattern of results is the same as when plotting ranked values. For the reasons outlined above, we believe the original versions of Figure 6 & Figure 7 with ranked values are actually easier to interpret and better represent the VLSM results they are designed to illustrate, particularly now complimented by the new (non-ranked) Figure S6 & Figure S7. However, we would be happy to also change Figure 6 & Figure 7 to the below if required:

6. The data and code are not available to reviewers. The depository is currently locked and requires a request for access.

Response: We apologise for not including an accessible link, this was an oversight on our part. We have now tested and included an updated review-only link and will provide a publicly accessible link on publication:

https://osf.io/xdnek/?view_only=6773e1653b744c0f846662eeb00c0a30

7. Why is the comparison of force exerted (Fig 5) only made between the vmPFC and healthy control group? The lesion controls should be included in this comparison.

Response: Apologies for any confusion, the analysis was indeed conducted between all 3 groups (Table S5 in original, now S6) and showed lesion controls were not significantly different from the vmPFC group in any aspect. The lesion controls also did not differ from healthy controls in how the recipient affected the force exerted. The original plot therefore highlighted the difference between the vmPFC group and healthy controls for clarity. We have now added a third panel to Figure 5 and Figure S2 showing the pattern of behaviour in lesion controls and included the relevant statistics in the legend:

Figure 5. Patients with vmPFC lesions exert less force for other relative to self, particularly when a large amount of effort is required. Patients with vmPFC damage exerted less force to obtain a reward for the other person than themselves (post-hoc contrast=0.04, SE=0.01, Z=3.62, $p<0.001$), whereas this was not the case for healthy control participants (post-hoc contrast=0.01, SE=0.01, Z=1.32, $p=0.19$; LMM group*recipient interaction $b=-0.09$ [-0.18, -0.01], $p=0.036$) or lesion control patients (post-hoc contrast=0.02, SE=0.01, Z=1.70, $p=0.09$; LMM group*recipient interaction $b=-0.05$ [-0.16, 0.05], $p=0.33$). The difference between vmPFC patients and healthy controls in energising actions to benefit other, relative to self, was particularly large when the level of effort required was greatest (LMM group*recipient*effort interaction $b=-0.03$ [-0.05, 0.00], $p=0.037$) and also when the reward available was smallest (LMM group*recipient*effort*reward interaction $b=0.05$ [0.03, 0.08], $p<0.001$; Figure S2). These interactions were not significant when comparing vmPFC and lesion control groups (Table S6). Asterisk or not significant (n.s) at the end of the line represents the recipient*effort interaction in each group. Asterisks above the lines represent significant differences ($p<0.05$) in post-hoc comparisons at each effort level.

Figure S2. vmPFC damage decreases the force exerted to gain rewards for another person, compared to oneself, as the effort required increases, particularly for the smallest rewards. In addition to the group*recipient and group*recipient*effort interactions shown in Figure 5, the linear mixed-effects model of force exerted showed a significant 4-way group*recipient*effort*reward interaction for vmPFC patients compared to healthy controls ($b=0.05$ [0.03, 0.08], $p<0.001$; Table S5). The corresponding interaction for vmPFC patients compared to lesion patients was not significant ($b=0.02$ [-0.01, 0.05], $p=0.17$). This sensitivity to the reward available and effort required, combined with high overall success rates, suggests vmPFC patients' reduced willingness to exert force for prosocial rewards was not due to an inability to meet the required force or attend to the information in the trial.

8. Please clarify if the post-hoc comparisons labeled in the figures with asterisks are $P<0.05$ after correcting for multiple comparisons.

Response: We have added this information to the figure legends:

Asterisks between self and other represent significant differences within each group in post-hoc comparisons ($p<0.05$ uncorrected: follow-ups of significant GLMM interactions)... Asterisks represent significant differences in post-hoc comparisons at

each effort level ($p < 0.05$ uncorrected: follow-ups of significant GLMM interactions).

9. The VLBM maps appear somewhat strangely constrained to the gray matter (e.g. Figure 8). Were the lesion masks in some way masked to exclude any WM voxels?

Response: Apologies for any confusion. The lesion masks were not constrained to the grey matter. We have provided an additional supplementary figure showing voxels included in the VLSM (Figure S4), please see our response to your minor comment 1 below.

Some more minor comments:

1. It would help to have a supplementary figure showing which voxels are actually being included in the VLBM analysis.

Response: Thank you for this suggestion. We have added it as a supplementary figure (Figure S4), reproduced below:

We only included voxels where at least five patients had damage⁶⁰ (Figure S4).

Voxels were only included in the VLSM if at least five patients had damage in that voxel to some extent^{45,60} (Figure S4).

Figure S4. Voxels included in voxel-based lesion-symptom mapping (VLSM) analysis where at least five patients had damage. We conducted VLSM once in all patients with damage ($n=45$) shown in darker purple and once limited to patients in the vmPFC group ($n=20$; see Methods).

2. Please include equations for the GLMM analyses in the methods section.

Response: Thank you for the suggestion. These have been added to the methods section in the supplement where the models are described:

The final models for each variable were:

Credits \sim Recipient*Group + (1|ID) with gamma log link function

Choice ~ Effort + Reward + Recipient + Group + Effort:Group + Reward:Recipient + Recipient:Group + Effort:Reward + (1 + Reward + Recipient + Effort + Reward:Recipient + Effort:Reward | ID) with binomial link function

$K \sim \text{Recipient} * \text{Group} + (1 | \text{ID})$ with gamma log link function

$\beta \sim \text{Recipient} * \text{Group} + (1 | \text{ID})$

Force ~ Effort*Recipient*Reward*Group + (1 + Effort + Recipient + Reward + Reward:Recipient | ID)

3. There are several references to cytoarchitectonic regions in the paper (e.g. 14m), but I cannot find an explanation or reference for how these areas were defined here. In general, I think it would be more helpful to have peak MNI coordinates for these effects.

Response: We agree further information would be helpful. We used the cytoarchitectonic maps of Öngür and Price (2000) as these provide empirical evidence for important divisions within sub- regions of more medial portions of vmPFC on the medial wall (area 14m that combines 14r and 14c) and more lateral portions of vmPFC on the orbital gyrus (areas 13a 13b and 13m). We also consulted the atlases provided by Neubert and colleagues (2015). We have added a supplementary figure (Figure S5) highlighting the distinction between these adjacent regions and their homology across species to support their anatomical labelling. We also now refer just to “area 14” and “area 13” throughout the paper (see response to your point 6). We agree that peak MNI co-ordinates would be very helpful and have also added these in the figure legends, thank you for this suggestion.

Architectonic map of the prefrontal cortex of macaques (Carmichael and Price, 1994)

Choice	Less prosocial
K	
Choice	More prosocial (vmPFC only)
K	

Architectonic map of the prefrontal cortex of humans (Öngür & Price, 2000)

Figure S5. Architectonic maps of the prefrontal cortex in humans and macaques showing divisions between area 14 (or 14m which comprises 14r and 14c) and area 13 (comprising 13a, 13b and 13m). Voxel-based lesion mapping showed two adjacent but distinct areas of vmPFC where motivation in patients with damage was either less prosocial (red / yellow) or relatively more prosocial (blue / green; in vmPFC patients only analysis). These areas correspond to separate cytoarchitectonic regions: area 14 on the medial wall (alternatively labelled 14m and comprising 14r and 14c; red) and area 13 on the orbital gyrus (comprising 13a, 13b and 13m, blue). These anatomical divisions are preserved across species^{11,12}.

Introduction: One possible explanation for conflicting findings is that vmPFC is a heterogeneous region comprised of several distinct cytoarchitectonic zones^{27,47-49}. Indeed, the term vmPFC is not an anatomical but a functional characterisation, and further understanding of the contribution of anatomical subregions within this heterogeneous zone is crucial. Whilst different characterisations of vmPFC subregions have been proposed^{47,49}, one prominent distinction is between medial and lateral areas^{47,48}. The most medial areas include area 14 on the medial wall (comprised of 14m, 14a and 14c) whereas more lateral areas include those on the orbital gyrus or area 13 (comprised of 13a, 13b and 13m). These distinctions could be key for understanding distinct functions⁵⁰⁻⁵³. So far, different functional contributions have been hypothesised including unique roles in representing positive and negative valence⁵³⁻⁵⁶, primary and secondary reinforcement⁵⁷, or processing identity^{58,59}, which is crucial for separating self and other. However, since focal vmPFC damage is so rare, most lesion studies are either case studies or involve fewer than ten patients¹³⁻²¹. Larger samples may permit lesion mapping approaches that can differentiate between these different vmPFC subregions, within the lesioned area. Lesion mapping can also reveal paradoxical positive and negative associations between behaviour and damage in close but distinct areas⁶⁰.

Results: "Lesions to regions putatively in area 14⁴⁹ (**Figure S5**) and area 25⁴⁸ (subgenual anterior cingulate cortex, sgACC) decreased willingness to help on both measures of choices (**Figure 6B**) and effort discounting (*K* parameters; **Figure 6C**)."

"Strikingly, this revealed a distinct, lateral portion of vmPFC, putatively in area 13⁴⁹ (**Figure S5**),"

Figure 6. Damage to medial portions of vmPFC specifically is associated with reduced prosocial behaviour. (a) Permutation-based, whole-brain, non-parametric voxel-based lesion-symptom mapping (VLSM) reveals subregions of vmPFC on the medial surface where damage is associated with reduced prosociality. This effect was found for both the self-other difference in choices to accept the work over the rest option (orange) and the self-other difference in devaluing rewards by effort (*K* parameters; red). Specifically, the recipient effect on choices was associated with damage to area 14 (peak 0, 30, -22; see **Figure S5**) and area 25 (subgenual anterior cingulate cortex, sgACC; peak ±2, 16, -6). The recipient effect on discounting *K* parameters was also associated with damage to area 14 (peak ±2, 14, -20) and area 25 (peak ±2, 14, -8). Plotting the (ranked) effect of the recipient on choices (b) and *K* parameters (c) separately for participants with damage or no damage in the areas identified by the relevant VLSM analysis reveals damage decreases prosociality. To further interpret these differences, we also plotted choices and *K* parameters for self and other separately by damage to the medial vmPFC regions (see **Figure S6**).

Figure 7. Lateral vmPFC damage is associated with relatively increased prosocial behaviour within the vmPFC patient group. (a) Permutation-based, whole-brain, non-parametric voxel-based lesion-symptom mapping (VLSM) shows damage to a lateral portion of vmPFC is associated with relatively increased prosociality (see **Figure S7** for comparison with average behaviour of healthy controls). An overlapping region in area 13 was identified in both the VLSM of the self-other difference in choices (dark blue; peak ±12, 20, -22) and how rewards were devalued by effort (*K* parameters;

teal; $\pm 14, 20, -12$; see **Figure S5**). Plotting the (ranked) self-other difference by whether participants had damage in the identified region shows the relatively increased prosociality in both choices (**b**) and K parameters (**c**), compared to patients with damage elsewhere in vmPFC. To further interpret these differences, we also plotted K_{self} and K_{other} separately (see **Figure S7**).

Figure 8. Damage to medial and lateral portions of vmPFC decreases sensitivity to reward.

Permutation-based, whole-brain, non-parametric voxel-based lesion-symptom mapping (VLSM) shows portions of vmPFC spanning medial and more lateral areas where damage is associated with reduced reward sensitivity. More damage in these subregions, areas 14, 13, 32 and sgACC, predicted a less positive effect of reward on choices in both the analysis of all patients (light brown; peak $6, 26, -2$) and the vmPFC group only (dark brown; peak $\pm 6, 24, -2$).

4. It would help if the individual participants were connected by lines (e.g Fig 3 a and b) to show the magnitude of the within-participant changes in behavior between the self and other conditions.

Response: Thank you for this helpful suggestion. We have updated all of the relevant plots to include within-participant lines (Figure 3A, Figure 3B, Figure 4D and Figure S1):

5. There seems to be a mistake on line 386: “As participants were generally less willing...”

Response: Thank you for spotting this, the line now reads:

“As participants were generally less willing to help someone else than themselves, this effect showed where damage was associated with even less willingness to help the other person relative to oneself.”

6. Aside from concerns with the second VLBM analysis, the idea of differentiating between more medial and lateral parts of vmPFC (an area with the word ‘medial in the name!) seems peculiar to me. I also am somewhat skeptical that such fine differences can be resolved in a lesion sample like this with large, messy and heterogeneous damage.

Response: Thank you for this query. Several studies have highlighted important divisions between medial (area 14) and lateral (area 13) parts of vmPFC across multiple species from rodents to non-human primates and humans (see response to comment 3 above), and these divisions are based on empirically determined cytoarchitecture (Öngür & Price, 2000). While these subdivisions only pertain to a secondary hypothesis, we believe it is valuable for a wider readership to highlight these differences as they also correspond to differences in behaviour in our sample. This is relatively unique compared to several existing vmPFC studies that either included samples with diffuse damage in addition to vmPFC or included samples with focal damage but fewer than 10 patients. However, we do agree that cytoarchitectonic boundaries can vary significantly across individuals and more precise techniques than lesion studies are critical for drawing strong conclusions about anatomical boundaries. We included some discussion to this effect in our original manuscript and we have now expanded on this section in line with your comments. We also now only refer to “area 14” and “area 13” given the difficulty of inferring precise anatomical boundaries between multiple sub-regions. This classification is supported by the new supplementary figure (Figure S5) and from the peak MNI coordinates now included in figure legends (also in response to minor point 3 above).

Architectonic map of the prefrontal cortex of macaques (Carmichael and Price, 1994)

Choice	Less prosocial
K	
Choice	More prosocial (vmPFC only)
K	

Architectonic map of the prefrontal cortex of humans (Öngür & Price, 2000)

Figure S5. Architectonic maps of the prefrontal cortex in humans and macaques showing divisions between area 14 (or 14m which comprises 14r and 14c) and area 13 (comprising 13a, 13b and 13m). Voxel-based lesion mapping showed two adjacent but distinct areas of vmPFC where motivation in patients with damage was either less prosocial (red / yellow) or relatively more prosocial (blue / green; in vmPFC patients only analysis). These areas correspond to separate cytoarchitectonic regions: area 14 on the medial wall (alternatively labelled 14m and comprising 14r and 14c; red) and area 13 on the orbital gyrus (comprising 13a, 13b and 13m, blue). These anatomical divisions are preserved across species^{11,12}.

Introduction: One possible explanation for conflicting findings is that vmPFC is a heterogeneous region comprised of several distinct cytoarchitectonic zones^{27,47-49}. Indeed, the term vmPFC is not an anatomical but a functional characterisation, and further understanding of the contribution of anatomical subregions within this heterogeneous zone is crucial. Whilst different characterisations of vmPFC subregions have been proposed^{47,49}, one prominent distinction is between medial and lateral areas^{47,48}. The most medial areas include area 14 on the medial wall (comprised of 14m, 14a and 14c) whereas more lateral areas include those on the orbital gyrus or area 13 (comprised of 13a, 13b and 13m). These distinctions could be key for understanding distinct functions⁵⁰⁻⁵³. So far, different functional contributions have been hypothesised including unique roles in representing positive and negative valence⁵³⁻⁵⁶, primary and secondary reinforcement⁵⁷, or processing identity^{58,59}, which is crucial for separating self and other. However, since focal vmPFC damage is so rare, most lesion studies are either case studies or involve fewer than ten patients¹³⁻²¹. Larger samples may permit lesion mapping approaches that can differentiate between these different vmPFC subregions, within the lesioned area. Lesion mapping can also reveal paradoxical positive and negative associations between behaviour and damage in close but distinct areas⁶⁰.

Discussion: Future work could compare the different functions subserved by areas 13 and 14 with electrophysiological recordings, to directly compare the properties of neurons in finer-grained medial vs. lateral vmPFC divisions. It is also important to note that the boundaries of different cytoarchitectonic areas vary significantly between individuals⁷⁰, and thus distinctions between these areas may only become apparent with even larger sample sizes. Additionally, separating these areas in samples with heterogeneous damage within vmPFC can be challenging. We do not interpret our results as suggesting a sharp, dividing line between these anatomical regions. However, we note that some differentiation of function between lateral and medial vmPFC is consistent with studies across different species and with cytoarchitectonic boundaries that have homologues across species^{47,48}.

Reviewer #2:

Remarks to the Author:

This study involves a rare sample of 25 ventromedial prefrontal cortex (vmPFC) lesion patients as well as non-vmPFC lesion patients and age/gender matched controls. The authors sought to determine the causal contribution of vmPFC to prosocial motivation, effort, and reward during decision making. The authors used neuroimaging, computational modelling, and a sound task design in an interesting and important study. Findings demonstrate vmPFC damage decreases prosociality such that vmPFC lesion patients earned less, were more reluctant to exert effort, and physically exerted less force when another person would benefit compared to the control groups. Reduced prosociality was related to medial damage with a paradoxical increase in prosocial motivation for lateral vmPFC lesions. The authors speculate based on prior literature that choice options for lateral area 13 is important for social decision making, but caution strong conclusions

that divide lateral and medial vmPFC with respect to functions in prosocial decision making.

Overall, this is a study with a strong design, unique sample, solid analysis, and worthy of publication in NHB.

Strengths: Task dissociates reward and effort, unlike economic games that conflate reward with financial cost. Large sample of vmPFC lesion patients. Control samples include matched non-lesion and non-vmPFC lesion patients. Comparison of self versus prosocial motivation highlights that vmPFC lesion is not disrupting motivation or reward processes generally but specifically prosocial motivations.

Response: Thank you so much for your positive feedback about our work. We are delighted that you appreciated our strong design, unique sample, and solid analysis. We really appreciate your time reviewing our work and your valuable comments, which have significantly helped us to improve our manuscript.

Minor suggestions: The introduction feels a bit disjointed. It might be better organized by moving the second paragraph up first and talking about the effort motivation followed by prosocial behaviour and finally the neural background.

Response: Thank you for this suggestion. We have now edited the introduction for greater clarity.

With the young adult / older adult discussion it would be helpful to tie this into the brain somehow – is the theory that the age trend is due to vmPFC maturation?

Response: Thank you for this suggestion. We indeed believe it is possible age-related differences in prosocial motivation may be underpinned by vmPFC maturation, although that was not the primary aim of the current study. We have now expanded our discussion to relate it to vmPFC maturation across the lifespan:

“Previous research has shown young adults are less willing to put in effort to help other people than they are to help themselves, particularly when the effort costs are high^{22,24,36,37}. They also exert less force into prosocial acts, although older adults become relatively more prosocial and exert equal force³⁶. Other work has tied differences in prosocial behaviour with age to vmPFC maturation³⁹ suggesting a possible neural basis to age-related difference in prosocial behaviour.”

There are several theories suggested for vmPFC involvement in decision making, including processing of self versus other which is interesting. It would be nice for this to be brought back up in the discussion to understand what the authors think underlie findings in this study.

Response: Thank you for this suggestion. We agree further discussion of the role of vmPFC in self vs other would be useful:

“The impact of vmPFC damage on prosocial but not self-benefitting decisions suggests that vmPFC is not processing identity, or simply self compared to other, but computations to decide to put in effort into actions that benefit others. Our results also highlight the importance of separating self and other processing experimentally to examine the unique contribution of different brain regions, and to highlight their role in processing effort and reward during decision-making for self, other or both.”

Thank you for the opportunity to review this work. I look forward to seeing it in publication.

Response: Thank you so much! We really value the positive feedback.

Reviewer #3:

Remarks to the Author:

In the manuscript entitled 'Human ventromedial prefrontal cortex is necessary for prosocial motivation', the authors propose to dissociate the contribution of vmPFC to prosocial behaviour (decisions causing real beneficial outcomes for another person), effort, and reward with a decision-making task that manipulates these factors independently. The used computational modeling to disentangle the role of these factors. A large group of patients with rare focal vmPFC lesions (n= 25) was compared to patients with lesions elsewhere (n=15), and healthy age and gender-matched controls (n=40). Participants chose either to rest, or to exert effort to gain rewards, for themselves or another person. They found that patients with lesions here earned less, were more reluctant to exert effort, and physically exerted less force when another person would benefit, compared to both control groups. Using voxel-based lesion mapping, they found that medial damage led to antisocial behaviour, while lateral damage was associated with increased prosocial motivation. Patients with vmPFC lesions also showed reduced sensitivity to effort but not reward overall.

The paper presents a nice design which enables to test participants on social and non-social versions of the same task. The analyses are nice and globally convincing. I nevertheless raise a number of points where clarifications or improvements could be necessary.

Response: Thank you so much for your positive feedback about our work. We are delighted that you appreciated our design and analyses and found our conclusions convincing. We really appreciate your time reviewing our work and your valuable comments, which have significantly helped us to improve our manuscript.

At the end of the introduction (line=110) it is stated that 'vmPFC patients earned less', while in the results (lines 153-155) it is said that they 'earned the same amount as lesion controls for themselves' and 'did not significantly differ from healthy controls'. This seems contradictory. Is it because line 110 speaks about the average earning (for self and other)? If yes, I think this is confusing and induces the reader to think that vmPFC patients have an impaired performance even in the self condition. Alternatively, I think I understood later on that they earned less 'when another person would benefit', as written at the end of the sentence. I think the sentence could become clearer if it started with 'When another person would benefit, vmPFC patients ...' Same situation in the Discussion section, line 458: 'that lesions to vmPFC lead to earning less money' - > 'that, when another person will benefit, lesions to vmPFC ...'. (These are just suggestions; The authors might disagree).

Response: Thank you for these suggestions. We agree these changes would add clarity and we have implemented them in the revised manuscript.

I don't understand how vmPFC patients could earn 'the same amount as lesion controls for themselves (ratio=1.00, ...' while Figure 3a shows a difference in earned credits for self between vmPFC patients (approx. 380) and lesion controls (approx. 390). Could the authors please explain?

Response: By the ‘same amount’ we intended to mean that there was no significant difference in amount earned between vmPFC patients and lesion controls. Indeed, there is evidence in support of the null that there is no difference. Not that the number was the same. The ratio of 1 is from a post-hoc test on the GLMM, rather than a simple comparison of the number of credits. We have revised this sentence for clarity:

“In contrast, vmPFC patients and lesion controls did not statistically differ in the amount they earned for themselves, with Bayesian evidence supporting the null (ratio=1.00, $p=0.99$, Bayes factor (BF_{01})=3.05; see Methods). Patients with vmPFC damage also did not significantly differ from healthy controls (ratio=0.95, $p=0.59$, $BF_{01}=2.10$).”

Figures 5 and S2 should also include lesion controls. This would help understand why 2 patients from the lesion control group earned less than 200 credits for other (Figure 3a) while accepting to work more for other more than 50% of the time (Figure 3b).

Response: We have added lesion controls to these figures, also as requested by Reviewer 1. You are correct that choosing to work and then not exerting the required force helps to explain why 2 patients from the lesion control group earned less than 200 credits for the other person.

Figure 5. Patients with vmPFC lesions exert less force for other relative to self, particularly when a large amount of effort is required. Patients with vmPFC damage exerted less force to obtain a reward for the other person than themselves (post-hoc contrast=0.04, SE=0.01, $Z=3.62$, $p<0.001$), whereas this was not the case for healthy control participants (post-hoc contrast=0.01, SE=0.01, $Z=1.32$, $p=0.19$; LMM group*recipient interaction $b=-0.09$ [-0.18, -0.01], $p=0.036$) or lesion control patients (post-hoc contrast=0.02, SE=0.01, $Z=1.70$, $p=0.09$; LMM group*recipient interaction $b=-0.05$ [-0.16, 0.05], $p=0.33$). The difference between vmPFC patients and healthy controls in energising actions to benefit other, relative to self, was particularly large when the level of effort required was greatest (LMM group*recipient*effort interaction $b=-0.03$ [-0.05, 0.00], $p=0.037$) and also when the reward available was smallest (LMM group*recipient*effort*reward interaction $b=0.05$ [0.03, 0.08], $p<0.001$; Figure S2). These interactions were not significant when comparing vmPFC and lesion control groups (Table S6). Asterisk or not significant (n.s) at the end of the line represents the recipient*effort interaction in each group. Asterisks above the lines represent significant differences ($p<0.05$) in post-hoc comparisons at each effort level.

Figure S2. vmPFC damage decreases the force exerted to gain rewards for another person, compared to oneself, as the effort required increases, particularly for the smallest rewards. In addition to the group*recipient and group*recipient*effort interactions shown in Figure 5, the linear mixed-effects model of force exerted showed a significant 4-way group*recipient*effort*reward interaction for vmPFC patients compared to healthy controls ($b=0.05$ [0.03, 0.08], $p<0.001$; Table S5). The corresponding interaction for vmPFC patients compared to lesion patients was not significant ($b=0.02$ [-0.01, 0.05], $p=0.17$). This sensitivity to the reward available and effort required, combined with high overall success rates, suggests vmPFC patients' reduced willingness to exert force for prosocial rewards was not due to an inability to meet the required force or attend to the information in the trial.

It would be great to add a black vertical line in Figure 7b and 7c (as well as in Figure S5a and S5b) to show where the healthy controls are. In terms of K difference, I take from Figure 6c that they would be around 18-19, thus probably non significantly different from patients with 0-rank damage to Area 13a/b. However, in terms of choice difference, I take from Figure 6b that they would be around 27-28, thus far away from patients with 0-rank damage to area 13a/b. Is it because these patients have a lesion elsewhere? If yes, where?

Response: We have edited the following figures as suggested, adding a horizontal line at the average score for healthy controls. These show for K values for both self and other, the healthy controls are similar to patients with area 13 damage and different to patients with zero damage in this area (but damage to other parts of vmPFC). Note that we have added lines to supplementary figures Figure S6 & Figure S7 but not Figure 6 & Figure 7. This is because these figures plot ranked values of participant-level random effects

from the GLMM models. Figure S6 & Figure S7 are to aid the interpretation of these main effects and are much improved by now showing where the healthy controls are.

In each of these figures, all participants with “no damage” have damage elsewhere. For the analysis of all patients (Figure 6 & Figure S6), this damage could be anywhere else in the brain. For the analysis limited to vmPFC patients (Figure 7 & Figure S7), the damage is elsewhere in vmPFC. We have edited the figure legends to clarify this.

Figure S6. Damage to medial vmPFC regions decreases willingness to exert effort for others and increases willingness to obtain self-benefitting rewards. To further interpret the voxel-based lesion symptom mapping (VLSM) finding of decreased prosociality with damage to portions of vmPFC on the medial wall (areas 14 and 25), we plotted each recipient separately to examine how damage affects choices for other **(a)** choices for self **(b)** discounting K parameters for other **(c)** and discounting K parameters for self **(d)**. We categorised participants by whether they had damage in the areas identified in the VLSM analysis for each effect (choices and K ; see Figure 6A). **Patients with no damage to the identified region had damage elsewhere in the brain.** These plots suggest that decreased prosociality shown by the recipient effects (other vs. self) in the VLSM are driven by both lower willingness to help the other person and higher willingness to work for oneself in patients with damage to these regions. **The dotted line represents the mean value for the healthy control group for comparison.**

Figure S7. Damage to lateral vmPFC compared to other vmPFC subregions leads to relative increases in willingness to help others. Our voxel-based lesion symptom mapping (VLSM) analysis identified a lateral portion of vmPFC in area 13 where damage was, in contrast, associated with **relatively** increased prosociality. To further interpret this finding, we extracted the extent of damage for each participant in the region identified in the VLSM analysis of the recipient effect (other vs. self) on K parameters (see Figure 7A). The corresponding effect on choices was associated with a smaller, overlapping region of area 13. Plotting damage in this region against discounting K parameter for other

(a) and discounting K parameter for self **(b)** separately shows the **relative** increase in relative prosociality is driven by both lower discounting (higher willingness to work) for other and higher discounting (lower willingness to work) to some extent for self. **The dotted line represents the mean value for the healthy control group for comparison.**

In the Methods section, lines 713-715, the authors should cite a reference in support of the following claim: 'Fitting the data across groups using this method provides the most conservative comparison and is more robust to the influence of outliers than single-step maximum likelihood estimation (MLE).' Moreover, can't such an iterative, hierarchical model fitting method constraint model parameters for different participants to be drawn from the same higher-level distribution? In other words, doesn't this bias outliers towards the mean of the distribution, and doesn't it pull non- outliers towards outliers?

Response: Thank you for this suggestion. We have added a reference to the above statement, which follows a very commonly used method in the field based on summary statistics (Daw, 2011, <https://github.com/sjgershm/mfit>). As Daw concludes "For most questions of interest, it is important to

treat model parameters as random effects, so as to enable statistical conclusions about the population from which the subjects were drawn. Given such a model, the easy way to estimate it is the summary statistics procedure, and since this is so simple, transparent, and fairly well behaved, we recommend it as a starting point. The more complex alternative — the fit of a full hierarchical model based on an approximation to Equation 5 — is better justified and seems a promising avenue for improvement.”

Indeed, the hierarchical approach is considered more robust to the influence of outliers. It also presumes that outliers form the extreme ends of a normal distribution and any parameters that are extreme reflect a failure of appropriate model fitting rather than reflecting the true behaviour of the participants. It is more conservative because, as you correctly say, different participants’ parameters are constrained to be drawn from a particular distribution, and therefore estimated interindividual differences are smaller. We have added additional information to the Methods section:

“Fitting the data across groups using this method provides the most conservative comparison and is more robust to the influence of outliers than single-step maximum likelihood estimation⁹¹. It is therefore recommended over single step methods, where it is possible to implement⁹¹.”

Didn't the fact that the role of the 'other participant' of the study was played by the authors' colleagues, and that they didn't do the task, constitute a case of deception for the real participants of the task? Were the latter informed afterwards? How did they react? From an ethical point of view, how was deception addressed if it was deception?

Response: *Yes, the study used mild deception in the design. Although there was really another person that participants were introduced to, as shown in Figure 2B, that person did not always receive additional monetary payment based on the participant’s decisions and was a confederate of the experimenter, for practical and logistical reasons with running a very complex study with rare patients with focal vmPFC damage. Our ethics approved the use of deception, which is standardly approved for psychology experiments, and we used a debriefing procedure to check that no participants reported disbelief in the deception. We have added further details to the manuscript.*

“All participants provided written informed consent and ethical approval, including for implementing mild deception, was obtained from the Oxford University Medical Sciences Inter Divisional Research Ethics Committee and National Health Service Health Research Authority Ethics Ref. 18/LO/2152.”

“This procedure used mild deception to minimise as much as possible the role of social preferences or reciprocity⁸³ in motivating prosocial behaviour.”

There is a mistake in Equation (a) Parabolic in the supplementary material: '(1' is missing after R(t).

Response: *Apologies. Thank you for spotting this. There were some formatting issues with how our equations were displayed. The correct equations are listed below and correspond to those also described in the Hartmann et al (2013) manuscript. We have updated these in our manuscript:*

(a) Parabolic: $SV_{(t)} = R_{(t)} - \&K * E_{(t)}^{\$*}$

(b) Linear: $SV_{(t)} = R_{(t)} - \&K * E_{(t)}^*$

(c) Hyperbolic: $SV_{(t)} = \frac{\%_{(t)}}{\&()^{*},(t)-}$

Since Figure 3b shows a couple of vmPFC patients which accept nearly 100% of high work options for other, could the authors investigate why they didn't find at least a couple values of beta parameter above 3 for vmPFC patients in the Other condition (like for healthy controls and lesion controls) (Figure S1)? Moreover, could the authors check whether participants around 50% accept in Figure 3b (no matter the group nor the condition) are associated with near-0 values of beta parameter in Figure S1? If not, could they check which participants get near-0 betas and why?

*Response: Thank you for this query. In fact, no vmPFC patients accepted 100% of offers for the other person, although some accepted almost all offers (97% and 96%). However, in those individuals, the few trials where they choose to rest over accepting the work offer were **higher** value (low effort / high reward). Therefore, the model estimates a **lower** beta as their decisions are more noisy. This idea, that the beta parameter does not simply reflect the number of offers accepted, but captures the noise in how closely value predicts decisions, also applies to participants who accepted around 50% of offers. Some of these participants would still have higher values of beta if they consistently accepted offers with above average value and rejected the 50% with lower values. In contrast, other participants who also accepted around 50% were inconsistent: the offers they accepted vs. rejected were not consistently high vs. low value, making their beta parameter near zero. We have added the following explanation to the methods section to help avoid any confusion and now refer to the beta parameters as decision "consistency" or "inverse stochasticity" throughout:*

"We quantified discounting of reward by effort (K) and decision consistency (inverse stochasticity β parameter) by comparing multiple models that represent different plausible theories of discounting. The β parameter is high if the participant consistently chooses the option with the highest subjective value, and low if there is noise in their decisions. Participants with low β values select to work or rest seemingly at random, irrespective of the subjective value of each option."

"from choice consistency (inverse stochasticity β parameter)"

Finally, it would be nice to cite people who have also tested parabolic (vs. linear vs. hyperbolic) discounting of reward by physical effort: Hartmann, M. N., Hager, O. M., Tobler, P. N., & Kaiser, S. (2013). Parabolic discounting of monetary rewards by physical effort. *Behavioural processes*, 100, 192-196.

Response: Apologies for omitting this important reference, we have now added it in the relevant places in the introduction and methods:

"Effort is aversive. If two courses of action are associated with different amounts of effort, but the same reward, people will choose the less effortful option²⁸⁻³²."

"We also varied whether the shape of the discount function was linear, hyperbolic, or parabolic³², creating a total of 12 models (see Supplementary Methods)"

Decision Letter, first revision:

15th January 2024

Dear Dr Lockwood,

Thank you once again for your revised manuscript, entitled "Human ventromedial prefrontal cortex is necessary for prosocial motivation," and for your patience during the re-review process.

Your manuscript has now been evaluated by the same reviewers who evaluated your original manuscript. All reviewer feedback is included at the end of this letter. Although the reviewers found your manuscript to have improved during revision, they also raise some important outstanding concerns. We remain very interested in the possibility of publishing your study in Nature Human Behaviour, but would like to consider your response to these outstanding concerns in the form of a revised manuscript before we make a decision on publication.

In specific, Reviewer 1 has some fundamental outstanding methodological concerns that we would like you to address in full.

In sum, we invite you to revise your manuscript taking into account all reviewer and editor comments. We are committed to providing a fair and constructive peer-review process. Do not hesitate to contact us if there are specific requests from the reviewers that you believe are technically impossible or unlikely to yield a meaningful outcome.

We hope to receive your revised manuscript within 4-8 weeks. I would be grateful if you could contact us as soon as possible if you foresee difficulties with meeting this target resubmission date.

- Include a "Response to the editors and reviewers" document detailing, point-by-point, how you addressed each editor and referee comment. If no action was taken to address a point, you must provide a compelling argument. This response will be used by the editors and reviewers to evaluate your revision.
- Highlight all changes made to your manuscript or provide us with a version that tracks changes.

[REDACTED]

We look forward to seeing the revised manuscript and thank you for the opportunity to review your work. Please do not hesitate to contact me if you have any questions or would like to discuss these revisions further.

Sincerely,

[REDACTED]

REVIEWER COMMENTS:

Reviewer #1:

Remarks to the Author:

Thanks to the authors for their efforts in revising this manuscript and providing a thorough response to comments and concerns. Many of my questions from the first round of review have been answered. However, I still have a few serious outstanding concerns that prevent me from recommending this manuscript for publication in NHB :

1. The authors have modified their VLBM analysis and now use the randomize function in FSL and TFCE thresholding to identify voxels associated with changes in behavior. However, it is not clear to me now how the behavioral measures are being inputted into this analysis – especially because this is not a common way to carry out this kind of analysis. I do not think that FSL alone would carry out the usual tests in VLBM (comparing patients with damage at each voxel to all other patients). The authors also rank the data and then carry out a parametric t-test on these ranks, which seems like an odd choice that is not explained. I would feel more comfortable if the authors were to use one of the existing software packages that is specifically designed for this purpose for this analysis (NiiStat, NPM, LESYMAP) without any additional transformations of the data (or please explain and justify these transformations). These software packages are easy to use and the analysis is very straightforward.

2. The demographic information the authors have provided indicate that the vmPFC group has considerably lower educational attainment compared to the healthy control group. The lesion control group also has significantly lower educational attainment, though the difference is not as stark. This is a major confound, as it suggests that the groups are not well matched – with the lesion groups likely being different on average in their premorbid cognitive function, as well as in their socioeconomic background and life experiences compared to the healthy controls. It seems plausible that educational attainment could have a major influence on the kind of social decision behavior.

3. I think that reporting that lateral ventral PFC damage is associated with ‘relatively’ increased prosociality in the abstract is fairly misleading. The authors do not find evidence that damage to this area results in a significant change in prosociality compared to controls. In Figure S7, some patients with damage to area 13 are more prosocial and some are less prosocial. On average, area 13 damage actually seems to be associated with less prosociality relative to the control group.

Reviewer #2:

Remarks to the Author:

The authors addressed my comments.

Reviewer #3:

Remarks to the Author:

The authors have addressed all my concerns.

Congratulations to the authors for this great work!

Mehdi Khamassi

Author Rebuttal, first revision:

Reviewer #1:

Remarks to the Author:

Thanks to the authors for their efforts in revising this manuscript and providing a thorough response to comments and concerns. Many of my questions from the first round of review have been answered. However, I still have a few serious outstanding concerns that prevent me from recommending this manuscript for publication in NHB :

Response: Thank you for your time re-reviewing our manuscript. We are delighted you appreciate our efforts in providing a thorough response to your comments and concerns. Thank you for your

additional feedback that has helped us to improve our manuscript further.

1. The authors have modified their VLBM analysis and now use the randomize function in FSL and TFCE thresholding to identify voxels associated with changes in behavior. However, it is not clear to me now how the behavioral measures are being inputted into this analysis – especially because this is not a common way to carry out this kind of analysis. I do not think that FSL alone would carry out the usual tests in VLBM (comparing patients with damage at each voxel to all other patients). The authors also rank the data and then carry out a parametric t-test on these ranks, which seems like an odd choice that is not explained. I would feel more comfortable if the authors were to use one of the existing software packages that is specifically designed for this purpose for this analysis (NiiStat, NPM, LESYMAP) without any additional transformations of the data (or please explain and justify these transformations). These software packages are easy to use and the analysis is very straightforward.

***Response:** We apologise for not including sufficient information about our updated VLBM analysis using FSL, particularly how the behavioural variables were put into the analysis. For the reasons we outline below, we believe that our analysis in FSL is justified, follows best practice, and is mathematically similar to the alternative software packages you suggest. We have now included more information about these justifications for our approach and how the analysis is run in our manuscript. We now also include instructions and code for running the analysis in FSL on our OSF page, in addition to the R scripts used to create the behavioural regressors and csv files containing the behavioural regressors that were shared alongside our first resubmission: https://osf.io/xdnek/?view_only=6773e1653b744c0f846662eeb00c0a30.*

FSL has been demonstrated to be valid to run VLBM analyses, and is commonly used, as evidenced by multiple recent lesion studies using this software (Cotovio et al., 2020; Pini et al., 2021; Trapp et al., 2023). We chose FSL as it is a robust analysis package, regularly updated by dedicated developers whilst remaining open source. It performs permutation-based analysis and implements the latest developments in neuroimaging analysis including threshold-free cluster enhancement (TFCE), a method that increases power and does not rely on defining thresholds which could be considered arbitrary. These criteria (non-arbitrary thresholding and permutation testing) were specifically highlighted as important in your previous comments. We agree they are indeed best practices and state-of-the-art and therefore re-ran all our VLBM analyses and implemented them. Our findings were robust across our original and updated VLBM analyses.

Critically, the approach we took, using FSL's randomise function, indeed compares patients with damage at each voxel to all other patients. Randomise performs permutation inference for the general linear model (Winkler, Ridgway, Webster, Smith, & Nichols, 2014):

$$y = \beta x + X$$

Our general linear model is:

$$(1) \text{ binary lesions} = \beta \times \text{behaviour} + X$$

As we only have a single regressor, this is exactly equivalent to:

$$(2) \text{ behaviour} = \alpha \times \text{binary lesions} + \chi$$

We use (1) because for permutation testing: “It is equivalent, however, to alter the design instead of the data... This is an important computational consideration as altering the design is much less burdensome than altering the image data” (Winkler et al. 2014).

From our understanding, the approaches and principles are the same in the other packages you highlight. The reference for NiiStat is the Winkler et al. 2014 paper that presents the randomise algorithm and their website states “For more details see the FSL randomise web page”. It also states “The statistical maps are typically identical to those generated with NPM (with minor rounding error differences)”. Similarly, the LESYMAP function that corresponds to our approach (“regresPerm”) fits GLMs to identify the relationship between brain and behaviour. Their website also cites Nichols & Holmes (2002) and Winkler et al. (2014) as the two papers outlining the mathematical principles. However, we note that NiiStat, NPM and LESYMAP, unlike FSL, have not been updated for several years and seemingly do not enable TFCE. The potential benefits of TFCE for VLBMs were highlighted by the LESYMAP developers (Mirman et al., 2018) but TFCE is not yet implemented in the software. We therefore suggest that FSL provides a more appropriate and up-to-date method for implementing the latest developments in VLBMs.

As you highlight, we do indeed rank the data then calculate *t*-statistics, but this is not an odd choice: it follows recommendations and standard practice. While permutation testing can be used to test the significance of any test statistic (parametric or non-parametric), it is preferable “to use a voxel statistic with approximately homogeneous null permutation distribution across the volume of interest, such as an appropriate *t*-statistic” (Nichols & Holmes, 2002). The developers of the randomise algorithm also recommend *t*-statistics as “appropriate for use with the general linear model and with a permutation framework, for being pivotal and easily implementable...” (Winkler et al. 2014). We rank the data to remove skew from the residuals, which is recommended practice (Winkler et al. 2014). The resulting *t*-statistic map shows where the behavioural regressor predicts binary lesions and vice versa: where patients with damage at that voxel differs from all other patients without damage. We then apply TFCE to the *t* maps to enhance power (Smith & Nichols, 2009) and test for significance through permutation testing.

Again, we apologise that our justification and details of our approach were not clear and appreciate you raising this point. In short, we have made the following changes, which have further improved the manuscript:

- (1) Included further information to detail the approach we used.
- (2) Added justification to our Methods section for using FSL with randomise and TFCE over other lesion mapping toolboxes.
- (3) Updated our OSF with the commands used for randomise and the behavioural regressor files for reproducibility.

Results: “Finally, we used voxel-based lesion-symptom mapping (VLSM) to examine whether specific subregions within the heterogeneous vmPFC were associated with prosocial behaviour, effort sensitivity, and reward sensitivity. The VLSM analysis identifies voxels where patients with damage at that voxel differ from patients with damage elsewhere in terms of self-other differentiation in (1) discounting (*K* parameters) and (2) choices, as well as how the effort required and reward available determined choices (see Methods).

Methods: “We quantified these variables for each participant by extracting the subject-level random effect of the relevant variable from adjusted GLMMs of choices and *K* parameters (*ranef* function; *lme4* package⁸⁷). These GLMMs included data from the five additional patients excluded from the group analysis (see above) and did not model any interactions or group as a predictor, as this would have accounted for some of the individual variance, which is the focus of VLSM. The subject-level random effect values were ranked to remove skew from the distribution of residuals⁶⁶

and z-scored, as required for the nature of our design, before being put into FMRIB software library (FSL)⁶⁴ design files. We then used FSL's *randomise* function to run permutation-based VLSM analysis^{65,66}, which compares patients with damage at each voxel to all other patients (see <https://osf.io/xdnek> for analysis code). FSL implements the latest developments in brain-based analyses, whilst remaining regularly updated and open source. FSL also enables the use of threshold-free cluster enhancement (TFCE) which maximises power and relies on non-arbitrary definitions of cluster size⁶⁵ and is not currently available in other lesion mapping toolboxes (e.g. LESYMAP, NiiStat). Patients' lesion maps were mirrored to increase power as we did not have hypotheses about laterality, creating symmetrical masks. Voxels were only included in the VLSM if at least five patients had damage in that voxel to some extent^{60,63} (Figure S4).

2. The demographic information the authors have provided indicate that the vmPFC group has considerably lower educational attainment compared to the healthy control group. The lesion control group also has significantly lower educational attainment, though the difference is not as stark. This is a major confound, as it suggests that the groups are not well matched – with the lesion groups likely being different on average in their premorbid cognitive function, as well as in their socioeconomic background and life experiences compared to the healthy controls. It seems plausible that educational attainment could have a major influence on the kind of social decision behavior.

Response: Thank you for your query regarding the demographic characteristics of our groups. Obtaining large samples of patients with focal lesion damage is non-trivial, as is testing older age-matched adults on a complex task involving multiple other people and bespoke equipment to measure force exertion. These practical aspects can impose some restrictions on our flexibility to completely match all groups on all measures. Our recruitment aimed to match all three groups on age and gender, which we achieved despite these challenges.

The vmPFC group was indeed lower in education than healthy controls. However, there were no differences between any pair of groups on two independent cognitive assessments of attention and executive function, Trail Making Test Part A and Part B. These tests assess participants' cognitive function at the time of the study, a more direct and relevant measure than inferring cognitive function from educational level, and show no difference between healthy controls and vmPFC patients. Additionally, our two lesion groups did not differ from each other in levels of education. We find the vmPFC group are less prosocial than both the lesion control and healthy control participants.

However, we appreciate the concern that differences in educational ability may still potentially impact our findings. To address your concern, we have now re-run all of our analyses controlling for level of education as a covariate. These multiple new analyses show no change in any of our key results or inferences regarding group differences in prosocial behaviour. We have updated all the relevant supplementary tables (Table S2 – Table S7, copied below) to show the new results, and added further information to the results section in the main manuscript where we report the demographic characteristics:

Results: “The three groups were carefully matched, with no differences in gender, age, cognitive ability, or levels of apathy. The two lesion groups also did not differ from each other in education or depression (see Methods and **Table S1**). However, the vmPFC and LC groups did differ from healthy controls in level of education. We therefore repeated all our behavioural analysis controlling for education as a covariate. Controlling for education did not change any of our key results or inferences regarding group differences in prosocial behaviour (see **Tables S2-S7**).”

New supplementary tables

Parameter	b	SE	CI low	CI high	t	p	LC n=20	Edu
(Intercept)	299.50	21.23	260.36	344.53	80.44	<0.001	<0.001	<0.001
Recipient (Self	1.25	0.04	1.17	1.33	6.59	<0.001	<0.001	<0.001
Group (vmPFC	1.20	0.11	1.01	1.44	2.03	0.044	0.10	0.12
Group (vmPFC	1.15	0.13	0.91	1.44	1.19	0.23	0.77	0.27
Recipient * Group	0.88	0.04	0.81	0.95	-3.07	0.003	0.001	0.003
Recipient * Group	0.87	0.05	0.78	0.97	-2.60	0.010	<0.001	0.011

Table S2. Generalised linear mixed-effects model predicting credits

vs. Other)

vs. HC)

vs. LC)

(vmPFC vs. HC)

(vmPFC vs. LC)

Note. HC: Healthy controls, LC: Lesion controls. LC n=20: supplementary analysis including the five additional patients with damage on the medial wall, but not focal to vmPFC, in the lesion control group (n=20). Results did not change compared to without these patients. Edu: supplementary analysis controlling for participants' levels of education by including this as a fixed effect in the model (main effect of education on credits $p=0.47$).

Parameter	OR	SE	CI low	CI high	z	p	LC n=20	Edu
(Intercept)	14.25	6.10	6.16	32.95	6.21	<0.001	<0.001	<0.001
Effort	0.40	0.08	0.27	0.60	-4.43	<0.001	<0.001	<0.001
Reward	6.91	1.36	4.70	10.15	9.83	<0.001	<0.001	<0.001
Recipient (Self	5.73	1.22	3.77	8.70	8.18	<0.001	<0.001	<0.001
Group (vmPFC	1.83	0.90	0.70	4.82	1.23	0.22	0.21	0.16
Group (vmPFC	2.09	1.32	0.60	7.21	1.16	0.25	0.35	0.23
Effort * Group	0.44	0.10	0.28	0.69	-3.57	<0.001	<0.001	<0.001
Effort * Group	0.73	0.21	0.42	1.27	-1.11	0.27	0.40	0.26
Reward * Recipient	1.71	0.14	1.46	2.01	6.56	<0.001	<0.001	<0.001
Recipient * Group	0.60	0.14	0.38	0.94	-2.22	0.026	0.022	0.044

Recipient * Group	0.48	0.14	0.27	0.84	-2.57	0.010	0.003	0.010
Effort * Reward	0.93	0.06	0.82	1.06	-1.06	0.29	0.27	0.35

Table S3. Generalised linear mixed-effects model predicting choices

vs. Other)

vs. HC)

vs. LC)

(vmPFC vs. HC)

(vmPFC vs. LC)

(Self vs. Other)

(vmPFC vs. HC)

(vmPFC vs. LC)

Note. HC: Healthy controls, LC: Lesion controls. LC n=20: supplementary analysis including the five additional patients with damage on the medial wall, but not focal to vmPFC, in the lesion control group (n=20). Results did not change compared to without these patients. Edu: supplementary analysis controlling for participants' levels of education by including this as a fixed effect in the model (main effect of education on choices \$p=0.51\$ ).

Parameter	b	SE	CI low	CI high	t	p	LC n=20	Edu
(Intercept)	0.08	0.02	0.05	0.12	-13.18	<0.001	<0.001	<0.001
Recipient (Self	0.54	0.04	0.46	0.64	-7.71	<0.001	<0.001	<0.001
Group (vmPFC	0.84	0.21	0.52	1.36	-0.72	0.47	0.48	0.35
Group (vmPFC	0.67	0.21	0.36	1.24	-1.28	0.20	0.30	0.17
Recipient * Group	1.36	0.13	1.12	1.65	3.17	0.002	0.002	0.002
Recipient * Group	1.46	0.18	1.15	1.85	3.13	0.002	<0.001	0.002

Table S4. Generalised linear mixed-effects model predicting K parameters

vs. Other)

vs. HC)

vs. LC)

(vmPFC vs. HC)

(vmPFC vs. LC)

Note. HC: Healthy controls, LC: Lesion controls. LC n=20: supplementary analysis including the five additional patients with damage on the medial wall, but not focal to vmPFC, in the lesion control group (n=20). Results did not change compared to without these patients. Edu: supplementary analysis controlling for participants' levels of education by including this as a fixed effect in the model (main effect of education on K parameters $p=0.50$).

Parameter	b	SE	CI low	CI high	t	p	LC n=20	Edu
(Intercept)	0.83	0.13	0.61	1.12	-1.24	0.22	0.27	0.57
Recipient (Self	1.69	0.16	1.40	2.05	5.41	<0.001	<0.001	<0.001
Group (vmPFC	1.36	0.27	0.92	2.01	1.56	0.12	0.12	0.54
Group (vmPFC	1.22	0.31	0.74	2.02	0.80	0.42	0.59	0.57
Recipient * Group	0.85	0.10	0.66	1.08	-1.35	0.18	0.18	0.18
Recipient * Group	0.84	0.13	0.61	1.15	-1.12	0.27	0.056	0.27

Table S5. Generalised linear mixed-effects model predicting β parameters

vs. Other)

vs. HC)

vs. LC)

(vmPFC vs. HC)

(vmPFC vs. LC)

Note. HC: Healthy controls, LC: Lesion controls. LC n=20: supplementary analysis including the five additional patients with damage on the medial wall, but not focal to vmPFC, in the lesion control group (n=20). Results did not change compared to without these patients. Edu: supplementary analysis controlling for participants' levels of education by including this as a fixed effect in the model (main effect of education on \$\beta\$ parameters \$p=0.057\$ ).

(vmPF Effort * Group vs. HC) Effort * Group vs. LC) Effort * Recipi vs. Other) Recipient * G (vmPFC vs. HC) Recipient * G (vmPFC vs. LC) Reward * Gro vs. HC) Reward * Gro vs. LC) Effort * Rewa Recipient * Reward	-0.01	0.01	-0.03	0.02	-0.53	0.59	0.54	0.61	
Effort * Rewa (vmPFC vs. HC) Effort * Rewa (vmPFC vs. LC) Effort * Recipi (vmPFC vs. HC) Effort * Recipi (vmPFC vs. LC) Effort * Recipi Reward									
Recipient * Reward * Group (vmPF Recipient * Reward * Group (vmPF				0.02	0.02	-0.01	0.05	1.55	0.12
Effort * Recipi Reward * Gro vs. HC) Effort * Recipi Reward * Gro				0.01	0.02	-0.03	0.04	0.34	0.73
					0.66	0.75			

vs. LC)

Note. HC: Healthy controls, LC: Lesion controls. LC n=20: supplementary analysis including the five additional patients with damage on the medial wall, but not focal to vmPFC, in the lesion control group (n=20). Results did not change compared to without these patients. Edu: supplementary analysis controlling for participants' levels of education by including this as a fixed effect in the model (main effect of education on force p=0.020).

Table S7. Linear mixed-effects model predicting success

Parameter	b	CI	t	p	LC	Edu			
SE	CI	CI	n=20						
low									
high									
	-0.30	0.16	-0.62	0.02	0.35 0.002	0.11 0.001	0.13 0.002	0.57	3.17
					0.54 0.009	0.21 0.059	0.14 0.085	0.95	2.63
					0.12	0.26 0.65	-0.40 0.63	0.64 0.79	0.46
Group	-0.32	0.14	-0.59	-0.04					
(Intercept) Recipient (Self vs. Other) Group									
(vmPFC vs. HC)					-0.18	0.18 0.30	-0.53 0.13	0.17 0.31	-1.03
Group (vmPFC vs. LC) Recipient * (vmPFC vs. HC)									
Recipient * Group									
(vmPFC vs. LC)									

Note. HC: Healthy controls, LC: Lesion controls. LC n=20: supplementary analysis including the five additional patients with damage on the medial wall, but not focal to vmPFC, in the lesion control group (n=20). Results did not change compared to without these patients. **Edu:** supplementary analysis controlling for participants' levels of education by including this as a fixed effect in the model (main effect of education on success $p=0.17$).

3. I think that reporting that lateral ventral PFC damage is associated with 'relatively' increased prosociality in the abstract is fairly misleading. The authors do not find evidence that damage to this area results in a significant change in prosociality compared to controls. In Figure S7, some patients with damage to area 13 are more prosocial and some are less prosocial. On average, area 13 damage actually seems to be associated with less prosociality relative to the control group.

Response: We are very sorry that including 'relatively' in our abstract came across as misleading, this was an oversight on our part. We have now made it clear that the comparison here is to patients with damage elsewhere, not to controls. [Please note we also edited the abstract more broadly for brevity to conform with the formatting changes requested by the journal editor for resubmission.]

"Whilst medial damage led to antisocial behaviour, lateral damage increased prosocial behaviour relative to patients with damage elsewhere."

Reviewer #2:

Remarks to the Author:

The authors addressed my comments.

Response: Thank you so much for your positive feedback about our work, we really appreciate your time and effort helping us to improve our manuscript.

Reviewer #3:

Remarks to the Author:

The authors have addressed all my concerns.

Congratulations to the authors for this great work!

Mehdi Khamassi

Response: Thank you so much for your positive feedback about our work and congratulations on our manuscript, we really appreciate your time and effort helping us to improve our manuscript.

Decision Letter, second revision:

18th March 2024

Dear Dr. Lockwood,

Thank you for your patience as we've prepared the guidelines for final submission of your Nature Human Behaviour manuscript, "Human ventromedial prefrontal cortex is necessary for prosocial motivation" (NATHUMBEHAV-23082573B). Please carefully follow the step-by-step instructions provided in the attached file, and add a response in each row of the table to indicate the changes that you have made. Please also address the additional marked-up edits we have proposed within the reporting summary. Ensuring that each point is addressed will help to ensure that your revised manuscript can be swiftly handed over to our production team.

We would hope to receive your revised paper, with all of the requested files and forms within two-three weeks. Please get in contact with us if you anticipate delays.

Nature Human Behaviour offers a Transparent Peer Review option for new original research manuscripts submitted after December 1st, 2019. As part of this initiative, we encourage our authors to support increased transparency into the peer review process by agreeing to have the reviewer comments, author rebuttal letters, and editorial decision letters published as a Supplementary item. When you submit your final files please clearly state in your cover letter whether or not you would like to participate in this initiative. Please note that failure to state your preference will result in delays in accepting your manuscript for publication.

In recognition of the time and expertise our reviewers provide to Nature Human Behaviour's editorial process, we would like to formally acknowledge their contribution to the external peer

review of your manuscript entitled "Human ventromedial prefrontal cortex is necessary for prosocial motivation". For those reviewers who give their assent, we will be publishing their names alongside the published article.

Cover suggestions

We welcome submissions of artwork for consideration for our cover. For more information, please see our guide for cover artwork.

ORCID

Non-corresponding authors do not have to link their ORCIDs but are encouraged to do so. Please note that it will not be possible to add/modify ORCIDs at proof. Thus, please let your co-authors know that if they wish to have their ORCID added to the paper they must follow the procedure described in the following link prior to acceptance:
<https://www.springernature.com/gp/researchers/orcid/orcid-for-nature-research>

Nature Human Behaviour has now transitioned to a unified Rights Collection system which will allow our Author Services team to quickly and easily collect the rights and permissions required to publish your work. Approximately 10 days after your paper is formally accepted, you will receive an email in providing you with a link to complete the grant of rights. If your paper is eligible for Open Access, our Author Services team will also be in touch regarding any additional information that may be required to arrange payment for your article.

Please note that *Nature Human Behaviour* is a Transformative Journal (TJ). Authors may publish their research with us through the traditional subscription access route or make their paper immediately open access through payment of an article-processing charge (APC). Authors will not be required to make a final decision about access to their article until it has been accepted. Find out more about Transformative Journals

[REDACTED]

Best regards,
[REDACTED]

On behalf of

[REDACTED]

Reviewer #1:

Remarks to the Author:

Thanks again to the authors for their work addressing my concerns. I appreciate the additional explanation for their VLBM methods and the additional analyses including education as a covariate.

I also appreciate the addition of more cautious language for the interpretation in the discussion in the last revision.

I would just lightly suggest to tone down the last sentence of the abstract a little given these caveats about how specific we can be about the functions of different vmPFC subregions with human lesions and VLBM.

Congrats on this work!

Author Rebuttal, first revision:

Reviewer #1 (Remarks to the Author):

Thanks again to the authors for their work addressing my concerns. I appreciate the additional explanation for their VLBM methods and the additional analyses including education as a covariate.

I also appreciate the addition of more cautious language for the interpretation in the discussion in the last revision.

I would just lightly suggest to tone down the last sentence of the abstract a little given these caveats about how specific we can be about the functions of different vmPFC subregions with human lesions and VLBM.

Congrats on this work!

Response: Thank you so much for your congratulations and for your 2me re-reviewing our manuscript. We are delighted that you appreciate the additional explanation for our VLBM methods and the additional analyses including education as a covariate. We have ensured the last sentence of our abstract does not mention vmPFC subregions:

"These results reveal multiple causal contributions of vmPFC to prosocial behaviour, effort, and reward."

Final Decision Letter:

Dear Professor Lockwood,

We are pleased to inform you that your Article "Human ventromedial prefrontal cortex is necessary for prosocial motivation", has now been accepted for publication in Nature Human Behaviour.

Please note that *Nature Human Behaviour* is a Transformative Journal (TJ). Authors may publish their research with us through the traditional subscription access route or make their paper immediately open access through payment of an article-processing charge (APC). Authors will not be required to

make a final decision about access to their article until it has been accepted. Find out more about Transformative Journals

With best regards,

[REDACTED]